# Microfluidic-Based Scratch Assays for Wound Healing Studies: A Systematic Review

**DOI:** 10.3390/cells14241931

**Published:** 2025-12-05

**Authors:** Fernando A. Oliveira, Nicole M. E. Valle, Keithy F. da Silva, Arielly H. Alves, Marta C. S. Galanciak, Gabriel M. Rosário, Javier B. Mamani, Mariana P. Nucci, Lionel F. Gamarra

**Affiliations:** 1Hospital Israelita Albert Einstein, São Paulo 05652-000, SP, Brazil; fernando.ao@einstein.br (F.A.O.); nicolemev@gmail.com (N.M.E.V.); keithyflx@gmail.com (K.F.d.S.); ariellydahora1997@gmail.com (A.H.A.); marta.caetano.2004@gmail.com (M.C.S.G.); gabrielmrosario16@gmail.com (G.M.R.); javierbm@einstein.br (J.B.M.); 2LIM44—Hospital das Clínicas da Faculdade Medicina da Universidade de São Paulo, São Paulo 05403-000, SP, Brazil; mariana.nucci@hc.fm.usp.br

**Keywords:** scratch assay, microfluidic device, wound-healing, scratch-on-a-chip, cell migration

## Abstract

Cell migration plays a central role in physiological processes such as wound healing, tissue regeneration, and immune responses, as well as in pathological conditions like chronic inflammation and tumor metastasis. Among the in vitro approaches to study this phenomenon, the conventional wound healing assay (scratch assay) has been widely used due to its simplicity and low cost. However, its limitations, including poor reproducibility, damage to the extracellular matrix (ECM), and lack of dynamic physiological conditions, have prompted the development of microfluidic alternatives. Scratch-on-a-chip platforms integrate engineering and microtechnology to provide standardized, non-destructive methods for wound generation, preserve ECM integrity, and allow precise control of the cellular microenvironment. These systems also enable miniaturization, reducing reagent and cell consumption, while facilitating the application of biochemical or physical stimuli and real-time monitoring. This review synthesizes advances reported in the literature, addressing the different wound induction strategies (enzymatic depletion, physical depletion, and physical exclusion), the role of ECM composition, and the impact of mechanical forces such as shear stress. Overall, scratch-on-a-chip assays emerge as promising tools that enhance reproducibility, better mimic in vivo conditions, and broaden applications for therapeutic testing and mechanistic studies in cell migration.

## 1. Introduction

Cell migration is a fundamental phenomenon in several physiological processes, such as wound healing, tissue regeneration, embryonic development, and immune responses, while also playing a central role in pathological conditions including chronic inflammation and tumor metastasis [1,2]. In wound healing, the coordinated movement of different cell types, such as keratinocytes, fibroblasts, and endothelial cells, is essential for the formation of new tissue. Therefore, analyzing this movement in controlled assays is crucial [1,3]. In this context, experimental methods have been developed to reproduce and monitor cell migration in vitro, enabling the investigation of its regulatory mechanisms and the design of therapeutic strategies that promote efficient regeneration under adverse clinical conditions [1,4].

Among the in vitro methods used to evaluate cell migration, the wound healing assay (scratch assay) stands out as the most conventional, low-cost, simple, and widely employed model in biomedical research [3,5]. In this assay, a confluent cell monolayer is mechanically injured, usually by scraping with the tip of a pipette, creating a cell-free area [6]. The advance of cells from the edges into the wounded region is then monitored over time, allowing the estimation of parameters such as migration rate and wound closure speed [1,6]. Despite its advantages, the conventional model presents limitations that compromise both reproducibility and the reliability of the results [5,7], such as the difficulty in standardizing wound width, which may damage or remove the extracellular matrix (ECM), restricted control of physicochemical variables, and the inability to reproduce dynamic conditions that mimic shear stress and physiological fluid flow of the natural cell microenvironment [4,8]. Furthermore, the release of intracellular contents from damaged cells can interfere with cellular responses to the microenvironment, and the static nature of the assay fails to reproduce physiologically relevant conditions such as fluid flow and shear forces found in vivo [7,9]. Thus, although it remains useful for screening and comparative experiments, there is a growing consensus on the need for more biomimetic models that realistically represent the cellular microenvironment.

To overcome these limitations, microfluidic devices have been developed for this assay as scratch-on-a-chip platforms, providing an alternative [7,10]. By incorporating principles of engineering and microtechnology, these systems enable the precise control of the cellular microenvironment, standardization of the wound area through non-destructive methods (such as removable physical barriers or enzymatic laminar flow), and preservation of the ECM [10]. In addition, they allow the miniaturization of experiments, reducing the consumption of reagents and cells, which is particularly advantageous in studies involving scarce or high-cost materials [11]. These platforms also make it possible to apply biochemical or physical stimuli in a controlled manner, monitor cell behavior in real time, and integrate with automated imaging systems [7,11].

Beyond offering greater experimental control, microfluidic platforms also allow the implementation of different methodologies for wound generation that are not feasible in conventional assays. Among them, there is enzymatic depletion, which uses controlled enzymatic digestion to locally remove cells; physical depletion, in which a localized mechanical force eliminates cells without compromising the surrounding area; and physical exclusion, which is based on removable barriers that initially prevent cell growth in a defined region, later creating the wound space once removed. Each of these strategies provides a standardized, reproducible, and less disruptive alternative to manual scratching, while better preserving the ECM and minimizing unwanted cellular damage [7].

However, limitations still restrict their widespread adoption, such as manufacturing complexity, the need for specialized instrumentation, a lack of methodological standardization, and a scarcity of correlation with conventional models, which also persist [12,13,14]. More complex models, such as skin-on-a-chip and microphysiological systems, already allow the integration of multiple cell layers and vascular networks, better reproducing tissue architecture and reducing the need for in vivo experimentation [15,16].

Despite these advances, the field still lacks comparative validation and quantitative physiological correlation with human data [12,13]. In this scenario, the integration between the rigor of classical methods and the precision of microfluidic systems is a promising strategy. Complementarily, the application of artificial intelligence (AI) can enhance the analysis of these systems. Machine learning and deep learning approaches already automate the detection of cellular patterns and predict biological responses in microfluidic models [17,18]. Although not yet directly applied to wound healing, these tools can assist in the precise quantification of migration and reduce experimental variability, representing a new analytical frontier to expand the standardization and translational value of these assays.

In this context, this systematic review presents a comprehensive and up-to-date analysis of the state of the art in microfluidic-based wound healing assays. Although previous reviews have contributed by exploring specific aspects of this topic, such as particular wound induction strategies, microfluidic principles, or delimited biological applications, a gap remains regarding a synthesis that integrates, in a structured way, all the steps involved, from the design and fabrication of the devices to their experimental application and physiological relevance. By adopting this integrated perspective, this work conducts an in-depth comparative analysis of the main technical, biological, and methodological advances, allowing the identification of trends, limitations, complementarities, and the applicability of different approaches in distinct cellular scenarios.

The review includes the characterization of microfluidic devices, their design principles and fabrication processes, cell culture conditions, lesion generation methodologies, and their respective experimental outcomes. Furthermore, it presents a critical evaluation of how design choices influence the experimental performance and physiological relevance of the models, relating each methodology to its biological context, the quantitative results observed, and its practical applicability. The work also introduces a timeline of technological evolution, compares costs and operational aspects between different microfluidic approaches and the conventional scratch assay, and discusses its translational applicability, including the potential for applications in personalized medicine and the main challenges involved in the clinical and regulatory adoption of these systems.

## 2. Materials and Methods

### 2.1. Search Strategy

This systematic review follows the Preferred Reporting Items for Systematic Reviews and Meta-Analyses (PRISMA) guidelines [19]. We searched indexed articles in PubMed, Scopus, and Web of Science. The following selected criteria of interest, keyword sequences ((“microfluidic” OR “microfluidics” OR “microfluidic device” OR “microdevice” OR “organ-on-a-chip” OR “lab-on-a-chip” OR “on-chip” OR “wound-on-a-chip” OR “microphysiological system”) AND (“scratch assay” OR “wound-healing” OR “wound-healing assay” OR “wound healing”)), and boolean operators (DecS/MeSH) used are shown in the Appendix A.

The present systematic review was prospectively registered in PROSPERO (International Prospective Register of Systematic Reviews) with the ID number CRD420251153369.

### 2.2. Inclusion Criteria

Eligibility criteria were established a priori. This review included only original articles written in English, published at any time up to February 2025. Studies were included if they (i) were in vitro studies, (ii) used microfluidic devices, and (iii) employed the scratch assay. These inclusion criteria were structured according to the PEO framework: Population (P)—in vitro cell cultures used for wound healing assays inside microfluidic devices; Exposure (E)—application of a wound induction method (enzymatic cell depletion, physical cell depletion, or physical cell exclusion) within the microdevice; and Outcome (O)—quantitative or qualitative assessment of wound closure, cell migration, or related healing parameters measured through microscopy, imaging analysis, or equivalent techniques.

### 2.3. Exclusion Criteria

Reasons for excluding studies were as follows: (i) review articles, (ii) book chapters, (iii) protocols, (iv) letters/communications, (v) conference abstracts or proceedings, (vi) publications in languages other than English, (vii) indexed articles published in more than one database (duplicates), (viii) studies that did not use microfluidic devices to perform the scratch assay, (ix) studies that performed wound healing assays exclusively in vivo or in vitro using non-microfluidic platforms, and (x) migration assays that did not apply the scratch method.

### 2.4. Data Collection

In this review, six of the authors (F.A.O., N.M.E.V., K.F.S., A.H.A., M.C.S.G., and M.P.N.) independently and randomly selected (in pairs), revised, and evaluated the titles and abstracts of the publications identified by the search strategy in the databases cited above (PubMed, Scopus, and Web of Science). All potentially relevant publications were retrieved in full. These same reviewers evaluated the full-text articles to determine whether the eligibility criteria were met. Discrepancies in study selection and data extraction between the two reviewers were discussed with a third independent reviewer (L.F.G.) and resolved.

### 2.5. Data Extraction

The selected articles were analyzed and summarized in dedicated tables addressing the following aspects: (1) Characteristics, design, and fabrication of microfluidic devices for scratch assays in wound healing studies; (2) Cell culture in microfluidic devices; (3) Wound healing assays in microfluidic devices; and (4) Proposal, evaluation, advantages, disadvantages, and outcome of wound healing scratch assays in microfluidic devices. The identification and extraction of data for each table were carried out by specific subgroups of authors, according to the focus of each aspect. F.A.O., N.M.E.V., K.F.S., and A.H.A. contributed to the analysis of studies involving microfluidic devices and cell culture methodologies. F.A.O., N.M.E.V., G.M.R., M.P.N., and J.B.M. focused on identifying and organizing studies related to scratch assay protocols and outcomes. The extraction of data and generation of the tables were performed by full consensus among the reviewing pairs. In case of discrepancies, a third independent reviewer (L.F.G.) resolved the issues. Final inclusion of studies in the systematic review was based on agreement among all reviewers.

### 2.6. Risk of Bias Assessment

The selection of articles was performed in 2 pairs and in case of disagreement, an independent senior author decided on the inclusion or not. The data selected in the tables followed the division of the authors by groups already described above and the checking of the data by the following group. In case of disagreement, author L.F.G. made the final decision.

### 2.7. Methodological Quality Assessment

The methodological rigor of the studies included in this review was evaluated using the Risk of Bias in Non-Randomized Studies of Interventions (ROBINS-I) tool, a well-established framework for assessing bias in non-randomized experimental research. This tool examines seven domains: confounding, participant selection, intervention classification, deviations from intended interventions, missing data, outcome measurement, and selective reporting of results [20]. Each domain is judged according to standardized criteria and categorized as Low, Moderate, Serious, Critical, or No information, depending on how effectively the study design and execution reduced the potential for bias [21]. The ROBINS-I evaluation was independently performed by three reviewers (N.M.E.V., K.F.S., A.H.A., and M.C.S.G.), all experienced in preclinical neurobiology. To measure inter-rater agreement, the Fleiss’ kappa coefficient was calculated, which indicates the level of concordance among multiple raters, with values ranging from 0 (no agreement) to 1 (perfect agreement) [22]. Calculations were carried out using JASP software (v0.19.3; https://jasp-stats.org/; accessed on 27 August 2025). Graphical summaries of the bias assessments were created with the robvis R package (version 0.3.0.900) and its associated Shiny web application [23].

### 2.8. Data Analysis

All results were described and presented using percentage distribution for all variables analyzed in the tables, to allow comparison with the scientific literature on the subject.

## 3. Results

### 3.1. Selection Process of the Articles Identified According to the PRISMA Guidelines

We searched publications indexed in PubMed, Scopus, and Web of Science, published at any time up to February 2025, and a total of 1398 articles were identified (317 in PubMed, 740 in Scopus, and 341 in Web of Science). From PubMed, 1 article was excluded for being published in a language other than English, 67 were reviews, 23 were book chapters, and 18 were letters, communications, conference abstracts, or protocols. Additionally, 187 articles were excluded for not performing scratch assays in microfluidic devices. Thus, 21 articles were included from this database. From Scopus, 6 articles were excluded due to language, 306 were duplicates of articles found in PubMed, 170 were reviews, 35 were book chapters, and 83 were letters, communications, conference abstracts, or protocols. Moreover, 133 studies were excluded because they did not perform scratch assays using microfluidic devices. In total, seven articles from Scopus were included. From Web of Science, 277 were duplicates in PubMed or Scopus, along with 40 reviews, 3 book chapters, and 10 letters, communications, conference abstracts, or protocols. Another 10 articles were excluded for not using microfluidic devices for the scratch assay. In total, one article from Web of Science was included in the review. Thus, only 29 unduplicated full-text articles [8,9,24,25,26,27,28,29,30,31,32,33,34,35,36,37,38,39,40,41,42,43,44,45,46,47,48,49] were included in this systematic review, 21 from PubMed, 7 from Scopus, and 1 from Web of Science, as shown in Figure 1.

The 29 selected studies were analyzed regarding microdevice geometry and fabrication, the cell culture type and methodology, wound healing characteristics and protocol, and reported outcomes. Due to the different scratch assays employed, each table was divided based on the three main types of assays used for wound healing analysis inside the device: 15 studies (45%) used enzymatic cell depletion [8,27,28,30,31,32,34,37,41,42,43,44,45,46,49], 13 studies (40%) used physical cell depletion [9,24,26,27,33,35,36,38,39,40,44,46,47], and 5 studies (15%) used physical cell exclusion [25,29,38,44,48]. It is important to note that four studies applied more than one technique: Moghadam et al. [46] and Gupta et al. [27] used both enzymatic cell depletion and physical cell depletion, while Yin et al. [44] and Sticker et al. [38] combined physical cell depletion with physical cell exclusion.

### 3.2. Methodological Quality Assessment Outcomes

The methodological quality of the 29 studies included in this review was assessed using the ROBINS-I tool, covering seven domains of bias (Figure 2). Overall, most studies showed a low to moderate risk of bias across the evaluated domains.

Regarding bias due to confounding (D1) and bias in participant selection (D2), the majority of studies were classified as low risk, indicating that the experimental designs adequately reduced the influence of external factors. Bias in the classification of interventions (D3) and bias due to deviations from intended interventions (D4) presented several cases with insufficient information, reflecting a lack of methodological detail in some articles.

Bias due to missing data (D5) was predominantly low, suggesting that most studies adequately reported experimental losses or failures during execution. However, in the domain concerning measurement of outcomes (D6), some studies were rated as serious risk, mainly due to the absence of standardized analysis methods or the use of quantification techniques without robust validation. As for bias in selection of the reported results (D7), moderate risk predominated, indicating potential selectivity in the presentation of findings.

When considering the overall risk of bias, the studies were mostly classified as low to moderate, reinforcing the reliability of the evidence while also highlighting the need for greater methodological rigor in aspects such as standardization of outcome measures, detailed description of procedures, and transparency in reporting results.

To assess the reliability of these judgments, Fleiss’ Kappa was calculated to determine inter-rater agreement across the three reviewers. The overall agreement was considerable (κ = 0.662, standard error = 0.107; 95% CI: 0.452–0.872).

### 3.3. Characteristics, Design, and Fabrication of Microfluidic Devices for Scratch Assays in Wound Healing Studies

The fabrication process of the microfluidic device used in the scratch assay, including information on the mold cast employed in production, the characteristics of the microfluidic device, as well as the device assembly procedure and the execution of in silico tests, can be found in Table 1. Among the 29 studies selected for this review, none used a commercial device; all microfluidic devices employed in the scratch assay were produced in-house. [8,9,24,25,26,27,28,29,30,31,32,33,34,35,36,37,38,39,40,41,42,43,44,45,46,47,48,49].

The majority of studies (20 out of 29, 69%) [8,24,25,26,28,30,32,33,34,37,41,42,43,44,45,46,47,48,49] fabricated their mold casts using photolithography. SU-8 negative photoresist was the predominant choice, employed by fourteen studies (48%) [8,26,28,30,33,37,41,42,44,45,46,47,48,49], but alternative photoresists were selected by six studies (21%): one study (3%) used TMMF S2045 negative photoresist [38], one study (3%) employed AZ 50XT positive photoresist [24], one study (3%) innovatively used the polymer poly(UA-co-IBA) as the resist material [34], one study (3%) applied photolithography but did not specify the photoresist employed [25], and two studies (7%) report using photolithography but failed to identify the mold material [32,43]. Three studies (10%) fabricated the mold using 3D printing [9,27,29], two of which reported photosensitive ink as the material [9,29], and one reported 3D-printed material [27]. Five studies (17%) did not apply a mold fabrication step in the chip production process [31,35,36,39,40]. Additionally, one study (3%) did not describe either the fabrication method or the mold material [44].

Regarding device fabrication, 23 studies (79%) employed soft lithography [8,9,24,25,26,28,29,30,32,33,34,37,38,41,42,43,44,45,46,47,48,49], three (10%) used CO_2_ laser processing [31,36,39], one (3%) applied laser micromachining [27], one (3%) used xerography [35], and one (3%) used a cutting plotter technology [40]. Polydimethylsiloxane (PDMS) was the most reported material, appearing in 24 studies (83%) [8,9,24,25,26,27,28,29,30,33,34,35,37,38,41,42,43,44,45,46,47,48,49], in 20 cases as the sole material, and in 4 combined with other substrates: polymethyl methacrylate (PMMA) and silk film [27], glass [35], dual-cure thermoset [38], or a LiNbO_3_ substrate [29]. Additionally, two studies (7%) used only PMMA [31,40], one combined PMMA with Teflon tape [39], and one used acrylic [36]. Only one study (3%) did not report the material used [32]. These results reflect the strong standardization observed in microfabrication processes, with soft lithography emerging as the dominant fabrication approach and PDMS remaining the material of choice [50]. Even though there is some variation in mold fabrication techniques and substrate combinations, the recurring use of photolithography-based molds and PDMS replicas reinforces the consolidation of soft lithography as the “gold standard” for the production of microfluidic chips for wound formation testing in the three methodologies found) [50,51].

Most microfluidic devices were fabricated using a single layer (43%) [8,26,28,32,34,37,41,42,43,44,47,48,49], but two-layer devices were also highly prevalent (37%) [24,25,27,29,30,33,36,40,44,45,46]. Constructions with three layers [27,38] or six layers [31,39] were each observed in 7% of the studies, while a smaller proportion was built with four layers [35] (3%) and seven layers [9] (3%), respectively. This distribution suggests a trade-off between simplicity and functional sophistication in device design. Single-layer devices were most frequently reported, likely due to their simpler and faster fabrication, though at the expense of functional complexity [52]. Two-layer chips offered a balanced trade-off between performance and fabrication effort, while devices with three or more layers were less common, reflecting greater technical challenges despite higher physiological relevance [52].

Most of the studies included in this review (69%) used glass as the cover material for the microfluidic devices [8,9,25,26,27,28,29,33,35,37,38,40,41,42,44,45,46,47,48,49], with two of these studies combining glass with PMMA [9,25]. Other materials were less common, such as culture dishes in six studies (21%) [24,34,36,39,43,44], one of which was coated with collagen [34]. One study (3%) employed a nano-patterned PDMS slab [30], while another (3%) used PMMA alone as the cover material [31]. Among the sealing methods used in microfluidic devices, oxygen plasma treatment of the cover was the most common, applied in 15 studies (55%) [8,25,26,29,30,35,37,41,44,45,46,47,48,49]. One study combined plasma treatment with screws (3%) [25]. Other sealing strategies included the use of double-sided tape (10%) [31,39,40], direct PDMS adhesion (7%) [24,28], super glue (3%) [27], heat sealing (3%) [38], and the adhesive properties of acrylic-based double-sided pressure-sensitive adhesive (3%) [36]. Five studies did not report the sealing method used [9,33,34,42,43], and only the study by Murell et al. failed to disclose both the sealing material and method [32].

In silico testing to assess fluid dynamics within the devices was reported in twelve studies (41%), with COMSOL Multiphysics being the predominant software used in nine of these [9,29,31,32,36,44,46,47]. Other computational tools included ANSYS Fluent [27], STAR-CCM+ [38], and CFD-ACE+ [39], each reported in 3% of the studies. Nevertheless, most studies (59%) did not conduct any computational simulations [8,24,25,26,28,30,33,34,35,37,40,41,42,43,45,48,49]. Despite its potential to optimize design parameters and reduce prototyping costs, in silico testing remains underutilized in wound healing microfluidic research. This limited likely adoption stems from the need for specialized expertise in computational modeling and the restricted accessibility of simulation software, which pose practical barriers to its routine implementation [53,54]. However, in silico validation was more frequently applied to physical cell depletion systems. This tendency may arise from the higher predictability of fluid dynamics and particle transport phenomena compared to complex biological responses. In such cases, computational modeling can accurately simulate shear forces and particle trajectories, providing meaningful optimization before experimental validation (Figure 6C) [54,55].

### 3.4. Cell Culture in Microfluidic Devices

Table 2 presents the distribution of the main characteristics related to the cells used in the scratch assay within microfluidic devices. It details the cell line and type employed, the cell concentration at the time of seeding, the culture medium and its supplements, the type of culture established (2D or 3D), the surface coating applied, the incubation time, and the seeding method adopted (using a pump, capillarity, or other techniques). These data highlight the diversity of approaches used to establish and maintain cell cultures during wound healing assays in the different devices analyzed.

Regarding the cells used for cultivation within the devices in wound healing assays, four studies reported more than one cell type and/or line. Among them, three performed direct comparisons by culturing different cell types in separate chips—one study compared endothelial cells with tumor epithelial cells [44], one compared three smooth muscle cell lines [42], while one evaluated three distinct conditions: non-tumor epithelial, tumor epithelial, and mesenchymal [48]. The remaining one study employed co-cultures, combining more than one cell line within the same device [24]. Considering all reported cell lines, fibroblasts were the most used, appearing in 11 studies [24,26,27,29,30,31,34,35,39,40,43]. The NIH/3T3 line was the most frequent (six times) [29,30,31,34,39,43], followed by L929 (two times) [27,40] and three other lines cited only once each (HFD [35], Balb/3T3 [26], and CCC-ESF-1 [24]). Endothelial cells were reported in eight studies, with HUVEC being the only line mentioned [24,37,38,41,44,47,49]. Epithelial cells also appeared in eight studies, of which five were tumor-derived (cervical adenocarcinoma [44], breast cancer [28], adenoid cystic carcinoma [48], melanoma [25], and prostate carcinoma [9]) and three non-tumor [8,32,33]. Two studies used microglial cells, all from the BV2 line [45,46]. Two studies employed keratinocytes (one HaCat [24] and one immortalized [36]). One study used smooth muscle cells with three different lines in the same work [42], and another used mesenchymal cells without specifying the line [48]. Overall, the cell-line distribution reflects a research focus on fibroblast-driven repair processes, with an emerging trend toward integrating endothelial and tumor cells to model angiogenic and oncogenic influences on wound healing. However, no clear association was observed between the type of cell line employed and the wound induction methodology used (Figure 6D).

In terms of cell concentration, 44.88% of the studies used concentrations ranging from 0.1 × 10^6^ to 4.5 × 10^6^ cells/mL [9,24,25,26,29,31,35,36,37,39,40,41,44], followed by 31.03% that applied concentrations greater than 4.5 × 10^6^ and up to 1 × 10^7^ cells/mL [8,30,34,42,43,45,46,48,49]. Concentrations above 1 × 10^7^ cells/mL were reported in 10.34% of the studies [28,33,44]. In two cases, the concentration was reported as the number of cells per chip, with 2 × 10^4^ in one case [47] and 3.75 × 10^6^ in the other [32]. Additionally, 6.90% of the studies did not report the concentration used [27,38]. Regarding the cell seeding method, 65.52% of the studies did not report how the cells were introduced into the chip [9,24,25,26,27,29,30,31,32,34,35,36,37,40,42,43,47,48,49], while 34.48% described using the hydrostatic passive method highlighting a preference for simplicity and manual control. Among these, five used pipettes [28,39,41,44,45], three used syringes [38,44,46], one used a needle [33], and one did not specify the tool employed [8].

Regarding the culture type, the two-dimensional (2D) strategy was widely employed, accounting for 97% of the studies [8,9,24,25,26,28,29,30,31,32,33,34,35,36,37,38,39,40,41,42,43,44,45,46,47,48,49], while only one study (3%) reported the use of three-dimensional (3D) models [27], which underscores the field’s early-stage focus on methodological validation rather than physiological complexity. In general, matrix elements were applied to coat the substrate, aiming to enhance cell adhesion. Concerning the coating type, the most frequently used was fibronectin, present in 38% of the studies [30,32,33,37,38,41,42,43,44,49], followed by collagen in 21% [25,33,34,35,42,45], and Poly-L-Lysine (PLL) in 10% [24,28,45]. These two coatings were preferred because they enhance adhesion and better reproduce extracellular matrix cues, reflecting efforts to increase physiological relevance, although the overall diversity of coatings indicates limited standardization across studies [56]. This preference was particularly evident among works employing enzymatic cell depletion, since these assays depend on controlled cell detachment, making the use of adhesion-promoting matrices essential to ensure reproducible wound formation (Figure 6E). Less common coatings included gelatin in 7% [38,45], and NIPAM [40], silk fibroin [27], and fibrinogen [38], each used in 3% of the studies. Eleven studies did not report matrix elements [8,9,26,29,31,36,39,44,46,47,48]. Finally, the incubation time of cells prior to forming the monolayer required for the scratch assay varied across studies.

### 3.5. Wound Healing Assays in Microfluidic Devices

In wound healing assays, the creation of the “wound” generally follows two main approaches: depletion and exclusion. In the selected studies, we identified three predominant strategies for generating this gap in the cell monolayer: enzymatic depletion, physical depletion, and physical exclusion. The details of wound formation within the chip, including both the method employed and the characteristics of the resulting wound, are summarized in Table 3.

Enzymatic depletion emerged as the most prevalent wound creation technique, employed by 15 studies [8,27,28,30,31,32,34,37,41,42,43,44,45,46,49]. This method involves the partial removal of a confluent cell monolayer through treatment with photolytic enzymes that facilitate localized cell detachment. All 15 studies used trypsin as the detachment agent. The introduction of trypsin solution varied across studies, with syringe pump being the most common delivery method (eight studies) [27,28,30,31,32,37,41,49], following by gravitational force (four studies) [34,42,43,46]. Other approaches included peristaltic pump [8], a manual pipette [45], and a passive pump with or without a siphon [44], each reported once (Figure 7A). This predominance likely reflects the simplicity and reproducibility of enzymatic protocols, which enable well-defined and controllable wound regions even in complex microchannel geometries. Regarding wound positioning, the majority of studies (eleven cases) [8,27,30,31,37,41,43,44,45,46,49] created a central region of the device, while four studies created it in a lateral region [28,32,34,42]. Regardless of location, all enzymatically generated wounds exhibited a linear geometry with varying dimensions [8,27,28,30,31,32,34,37,41,42,43,44,45,46,49]. The preference for linear shapes aligns with conventional macroscopic assays and facilitates the quantification of wound closure by straightforward imaging and analysis methods.

Physical depletion uses various thermal [40], magnetic [44], or mechanical [9,24,26,27,33,35,36,38,39,46,47] principles to detach cells from a confluent monolayer, thereby generating the wound. Thirteen studies employed this methodology [9,24,26,27,33,35,36,38,39,40,44,46,47]. Two of these studies performed both enzymatic and physical depletion, using different chips for comparison purposes [27,46]. Unlike the enzymatic approach, physical depletion provides flexibility in design and allows the integration of external actuation (e.g., pressure, heat, or magnetic stimuli), but at the cost of higher variability and lower reproducibility across platforms. The most frequent was air pressure, reported in three studies [33,35,38]. Other methods included Parafilm M^®^ [9], phosphate-buffered saline (PBS) [46], a pipette tip with a vacuum aspirator [36], a magnet-module system [44], a microrobot and micropipette [47], a PDMS mold [27], a rigid plastic piece with posts [26], thermoresponsive microgels [40], a PDMS stencil [24], and a tape-made barrier [39], each reported in one study (Figure 7B). The spatial positioning of wounds showed a clear preference for the central region of the device, with twelve studies [9,24,26,27,33,35,36,38,40,44,46,47], while one study did not report the wound location [39]. The geometric configuration of these wounds demonstrated considerable diversity across studies. A linear shape appeared in seven studies [9,24,27,36,39,46,47], followed by circular wounds in six studies [26,33,35,38,40,44]. Two studies employed a square shape wound [27,47], and in one study used a triangular configuration [47]. Notably, wounds varied in size across the different studies. The diversity of geometries—linear, circular, square, and even triangular—reflects the experimental versatility of physical methods, though it also underscores the lack of methodological standardization in this subgroup.

Physical exclusion involves preventing cell adhesion in specific regions of the substrate by placing physical obstacles that are later removed, leaving a cell-free area that forms the wound. Only five studies employed physical exclusion as the method for wound creation [25,29,38,44,48]. Two of these studies also applied physical depletion in a separate chip for comparison purposes [38,44]. In three studies, PDMS pillars were used to generate the wound [25,29,48]. Sticker et al. study [38] related a microstencil spin-coated PDMS as the wound generator [38], while one study utilized a system of pillars combined with a magnet-module system [44] (Figure 7C). This approach, though less common, avoids chemical or mechanical stress to cells and thus better preserves monolayer integrity prior to wound initiation, but it is experimentally more complex and time-consuming. In all these cases, the wound was created in the central region of the device. In two studies, the wound exhibited a linear shape [25,29], while in the other two it was circular [44,48]. The duration of obstacle placement varied between devices, with one study keeping the obstacle overnight [48], another for 6 h [44], and another for 8 h [29]. Only the study by Gao et al. did not report the duration of obstacle placement [25]. In the study by Sticker et al., the exclusion-based method using pneumatically removable stamps proved unfeasible, as the stencil hindered reproducible cell seeding; consequently, no wound formation results were obtained [38].

Representative examples of the methodologies used for wound formation in microfluidic platforms are illustrated in Figure 3, Figure 4 and Figure 5, showcasing the diversity of strategies reported across the analyzed studies. As shown in Figure 3, enzymatic depletion devices tend to adopt simple and highly similar geometries, generally composed of parallel linear channels that enable controlled enzymatic exposure and reproducible wound boundaries. In contrast, Figure 4 highlights the greater architectural variability of physical depletion systems, which employ pneumatic actuators, micropillar arrays, or robotic tools to generate wounds of customizable shapes while integrating mechanobiological readouts. Figure 5 depicts physical exclusion approaches, characterized by more intricate microarchitectures that prevent cell attachment through acoustic waves, removable pillars, or magnetically actuated stamps, allowing precise, non-traumatic wound delimitation ideal for preserving ECM integrity. Complementarily, Figure 6 and Figure 7 provide a graphical synthesis of the main findings, including methodological distribution, study objectives, and evaluation parameters.

### 3.6. Proposal, Evaluation, Advantages, Disadvantages, and Outcome of Wound Healing Scratch Assays in Microfluidic Devices

Appendix A summarizes the main objectives of each study in relation to the wound healing assay, along with the methods used for wound evaluation, the reported outcomes, and the advantages and limitations highlighted by the authors.

The research objectives across the reviewed studies revealed diverse focal areas in wound healing investigation. The largest proportion of studies (13 studies, 44%) [8,9,24,27,28,31,32,33,34,35,42,45,48,49] concentrated on evaluating the effects of chemical or biochemical factors on wound closure dynamics, examining how various molecular agents influence cellular migration and tissue regeneration. This focus reflects the strong interest in using microfluidic systems as controllable in vitro environments for screening bioactive compounds under dynamic conditions, an advantage compared to static culture models. Mechanical forces represented another prominent focus, with six studies (19%) [25,26,28,37,39,41] investigating shear stress effects on wound healing processes and cell migration patterns. Such emphasis highlights the recognition of biomechanical cues as critical regulators of wound repair, and the capacity of microfluidics to precisely modulate these stimuli.

Five studies (15%) [31,40,43,44,47] focused on methodological development, establishing reproducible protocols for creating consistent wound models in microfluidic platforms. These works reinforce the current effort toward protocol standardization, which remains a challenge across different device designs and materials. Three studies (10%) [29,30,46] examined how different extracellular matrix (ECM) compositions applied to cell cultures influence wound closure kinetics, emphasizing the relevance of matrix–cell interactions in recreating physiologically meaningful healing responses. Additionally, three studies (12%) [36,38,44,47] explored the therapeutic potential of physical stimulation modalities, investigating how electrical stimuli or acoustic waves affect cellular responses and wound closure processes. Although less frequent, this trend points to the growing integration of multimodal stimulation technologies in organ-on-chip and regenerative medicine research.

Wound closure was assessed using optical microscopy in 27 studies [8,9,24,25,26,27,28,29,30,31,32,33,34,35,36,37,39,40,41,42,43,44,45,46,47,48]. Fluorescence microscopy was employed in fifteen studies [8,24,25,29,30,32,33,35,37,38,40,41,44,48,49], and in two of these it was the only technique used [38,49]. Only one study used transmission electron microscopy (TEM) [8] (Figure 7D). The predominance of optical and fluorescence imaging underscores the reliance on conventional yet reliable visualization methods, which provide sufficient spatial resolution for quantifying closure dynamics without requiring complex instrumentation. The software used for wound area quantification varied among studies. A total of nineteen studies employed ImageJ [8,9,24,25,30,31,32,35,36,37,38,39,41,42,44,45,46,47]—sometimes in combination with Imaris [32], Matlab [32], CellTracker [36], or Adobe Photoshop [38]—followed by Image-Pro Plus in four studies [24,26,28,43], Fiji in three studies [29,44,49], MATLAB in one study [33], and Cell-R in one study [40]. Only two studies did not report the software used [34,48]. The predominance of open-source tools such as ImageJ and Fiji reflects a preference for accessible and customizable image analysis pipelines, although differences in software use may contribute to variability in data processing and comparability across studies.

Regarding monitoring duration, 14% of the studies evaluated up to 6 h [8,35,45,47], another 14% up to 12 h [25,37,38,44], 10% up to 18 h [27,32,44], and 38% up to 24 h [24,26,28,29,30,31,34,36,39,41,43]. In 10% of the cases, evaluation extended to 36 h [33,40,46], in 7% up to 48 h [9,48], and only one study reached 72 h [49] or 192 h [42] (Figure 7E). These timeframes indicate that most assays were designed to capture early migratory behavior rather than full tissue regeneration, aligning with the short-term dynamics typical of in vitro wound models.

Overall, the most frequently reported advantage among the studies included in this review was the low cost of the devices, along with their ability to minimize the use of reagents and consumables [24,25,29,30,32,35,37,47,48,49]. On the other hand, the main disadvantage observed was the requirement for specialized equipment to manufacture the microfluidic devices [32,33,34,35,36,38,40,43,44,45,46]. Nonetheless, all reviewed studies successfully employed the proposed microfluidic systems to evaluate wound closure, confirming the robustness and adaptability of these platforms across diverse experimental objectives and technical configurations.

### 3.7. Technological Evolution of Microfluidic Wound Healing Assays

Owing to the diversity of applications, fabrication strategies, and analytical approaches observed across studies, a temporal comparative analysis was conducted to trace how microfluidic wound healing assays have evolved over time. This perspective reveals a clear technological trajectory—from early, simple linear systems to increasingly complex, integrated, and physiologically relevant platforms (Figure 8).

Between 2007 and 2010, pioneering studies established proof-of-concept platforms employing flow-based enzymatic depletion [34,41]. These early systems introduced controlled laminar flow to reproducibly generate wounds and enabled the first quantitative assessments of cell migration under stable chemical gradients and shear stress.

From 2011 to 2013, research expanded to explore mechanical stress, peripheral cell damage, and electrical stimulation effects [8,32,39]. The introduction of electrical and mechanical cues into microfluidic environments provided new insights into cellular motility and wound repair, moving beyond passive observation to active control of the healing process.

Between 2014 and 2016, hybrid approaches combined microfluidic precision with traditional scratch assays [24,33,39,40]. Devices featuring thermoresponsive, non-destructive surfaces and cost-efficient fabrication processes emerged, bridging the gap between complex engineering designs and biological reproducibility.

From 2017 to 2019, platforms evolved toward reproducibility and structural sophistication [26,30,31,37,38]. Systems with pneumatic valves, patterned microtopographies, and integrated gradients allowed spatiotemporal control over wound geometry, while oxygen and nutrient gradients brought the models closer to physiological conditions.

Between 2020 and 2022, innovations introduced acoustic manipulation, microrobotics, and co-culture models [29,35,44,47,49]. These technologies enabled a deeper understanding of biomechanical regulation and paracrine communication in wound closure, extending microfluidic applications to multi-cellular and neuroimmune systems.

Finally, from 2023 to 2025, research focused on accessibility and translational potential [9,36,45,46]. The development of pump-free, low-cost, and standardized devices improved experimental scalability, while new designs integrating biomechanical–biochemical synergy analysis established a bridge between microengineering and regenerative medicine.

Overall, the timeline in Figure 8 illustrates how microfluidic wound healing assays have evolved from fundamental prototypes into robust, biomimetic, and cost-effective tools for quantitative wound biology, with increasing relevance to preclinical and translational research.

## 4. Discussion

This systematic review revealed that, to date, microfluidic-based wound healing assays remain an emerging approach, with only 29 studies identified in the literature. While the conventional scratch assay is a well-established and standardized method for assessing cell migration in vitro [6], this is not yet the case for microfluidic platforms, where considerable methodological heterogeneity was observed across studies. Although the conventional assay is still widely used for its simplicity and standardization [1,6], its limitation in reproducing dynamic in vivo conditions such as fluid flow and chemical gradients has driven the development of lab-on-a-chip platforms [8,24,31,41]. Radar plots comparing these approaches highlight that microfluidic models (Figure 9), particularly those based on physical and exclusion cell depletion, outperform the traditional scratch assay in terms of physiological fidelity, reproducibility, and integration of biochemical and mechanical gradients. Conversely, enzymatic depletion systems show intermediate performance—offering improved reproducibility and gradient control, yet at the expense of ECM preservation.

Consistent with the observations of Deal et al. (2020) [12] and Cho et al. (2024) [15], microfluidics emerges as an essential tool to model the wound microenvironment, providing precise control over shear stress, concentration of growth factors, and wound geometry, while also enabling integration with biosensors and automated monitoring systems [6]. Variations encompassed device design and architecture, wound induction methods, flow control, and cell culture conditions [2]. This lack of standardization represents one of the main obstacles to broader adoption of these models in comparative or translational research. In contrast, the fabrication of the microfluidic devices themselves appears to be well standardized, with consistent use of reproducible materials and fabrication protocols—reflected in this review by the finding that 79% of studies employed soft lithography, a mature, low-cost, and reliable technique enabling precise replication of microchannel geometries across laboratories [38,40,57]. The discrepancy between the technical reproducibility of device fabrication and the biological variability of wound healing assays highlights a central gap in the field: while fabrication materials and processes appear well established, the design and experimental application of microfluidic wound healing systems remain highly diverse, lacking clear consensus on device architecture or wound-generation strategies [58].

All studies included in this review employed in-house fabricated microfluidic devices, reflecting the current limited availability of commercial systems specifically designed for wound healing applications [59]. Although this reliance on custom-built platforms may constrain reproducibility and accessibility across laboratories, it also underscores the importance of flexibility and customization in this research field. The ability to tailor device geometry, flow configuration, and biological microenvironment allows researchers to adapt experimental conditions to specific cell types and scientific questions—advantages rarely achievable with standardized commercial chips [57,59]. Furthermore, the market for such systems remains in its infancy, with few available options that are typically expensive and limited in adaptability. Consequently, in-house fabrication remains a technical and scientific necessity, supporting the development of physiologically relevant and translationally oriented models. As microfabrication tools become more accessible and protocols more unified, greater standardization and scalability are expected to emerge in the near future [57,59,60].

Therefore, although the predominance of custom devices may initially limit generalization, it simultaneously drives advances in design, scalability, and standardization, progressively bringing microfluidic technology closer to translational and regulatory applications [7,25,46]. With the increasing standardization of PDMS and thermoplastic commercial chips and the adoption of low-cost additive manufacturing, the per-assay cost tends to approach that of traditional methods while maintaining superior control, scalability, and reliability [8,30]. Thus, although the scratch assay remains useful for basic applications, microfluidic platforms emerge as sustainable and economically advantageous alternatives, aligned with current demands for standardization, automation, and scientific efficiency.

This review showed that the main methodologies for wound creation in microfluidic devices were enzymatic cell depletion, physical cell depletion, and exclusion cell depletion. Enzyme depletion-based models are the most established among microfluidic wound healing assays and the most frequently reported in this review. In these models, the wound region is generated by laminar flow containing trypsin or EDTA, selectively removing cells in delimited areas without affecting adjacent ones [8,27,28,30,31,32,34,37,41,42,43,44,45,46,49].

Importantly, the geometry and architecture of the device are closely related to these methodological outcomes and should be chosen according to the wound formation strategy, experimental duration, and available infrastructure. Enzymatic depletion is usually implemented in simple, rectilinear channels that support laminar flow and stable enzyme gradients, conditions that ensure high reproducibility and controlled depletion while facilitating long-term use and easier fluid handling [7,12,31]. However, such simplicity limits experimental throughput and geometric variability, making these designs less suitable for studies exploring diverse wound morphologies or high-content analyses [25,46]. Compared to the scratch assay, the enzymatic method offers better spatial control and less mechanical damage, avoiding ECM removal and reducing non-physiological inflammatory response [10,12,31,61,62]. Furthermore, it allows the creation of stable chemical gradients under laminar flow, enabling the study of chemotaxis induced by growth factors such as VEGF, FGF-2, and PDGF-BB [8,27,34,49]. The use of passive pumps or gravitational systems simplifies operation while maintaining flow precision [42,45]. However, small variations in trypsin exposure time can affect wound width and cell viability [31,41,45]. These systems generally produce only one linear wound per chip, requiring multiple devices for high throughput [28,31,34]. One proposed alternative was the creation of regions of different sizes within the same microfluidic channel; however, this strategy achieved only three distinct areas, highlighting the need for further adaptations to increase experimental throughput [31].

Biologically, enzymatic depletion is ideal for investigating individual migration and response to soluble gradients, being widely used to evaluate closure kinetics, directional persistence, and cell edge reorganization [8,28,31,32,34,42]. The chip design, channel width, substrate (collagen or fibronectin), and gradient stability influence the speed and direction of migration [30,32,34,37,41,42,43,44,46,49]. Recent studies have demonstrated a correlation between laminar shear stress and the closure rate, highlighting the importance of integrating flow parameters into quantitative analysis [10,12,31,44,49,58,62]. Thus, the technique represents a physiological evolution of the scratch assay, although still limited by low parallelization and a lack of interlaboratory standardization.

Although physical depletion has advanced the control of mechanical and electrical forces in wound formation, it still relies on the direct rupture of the cell monolayer. Unlike enzymatic depletion, which depends on the biochemical detachment of cells through trypsinization, physical depletion generates wounds through mechanical, thermal, or magnetic disruption, enabling the study of the effects of externally applied forces on cell migration and wound repair dynamics. Devices based on pneumatic valves, microneedles, or microactuators generate wounds with a well-defined area and high reproducibility while preserving the surrounding ECM [9,26,33,35,38,47]. The use of electric fields in microfluidic chips has also proven effective in directing cell migration and accelerating wound closure, simulating the endogenous potentials present in injured tissues [7,36,39,62]. These models allow the study of mechanotransduction, that is, the cellular response to deformations and shear gradients, with precise control of the intensity and duration of exposure [10,24,27,38,45,62]. This feature represents a major improvement over enzymatic methods, which can cause heterogeneous detachment and potential damage to the ECM.

Compared to the conventional scratch assay, physical depletion maintains the traumatic nature of wound creation but eliminates uncontrolled operator variability and allows for greater reproducibility of wound geometry [10,26,35,36,38,39,47,62]. In addition, microfluidic implementation of this technique enables temporal and parametric quantification of migration processes, correlating applied force, cellular deformation, and cytoskeletal reorganization under controlled shear stress [10,24,27,38,39,45,47,62]. From a biological perspective, physical depletion is particularly useful for investigating mechanotransduction phenomena, such as cell polarization, actin alignment, and coordinated responses to mechanical stress [10,24,27,38,39,45,47,62]. However, despite its precision, excessive exposure to mechanical or electrical stimuli can cause peripheral necrosis or alter cell viability. Moreover, the use of multilayer configurations and pressurized systems increases fabrication complexity and operational cost, potentially limiting the method’s applicability in simpler experimental setups. Even so, physical depletion represents a bridge between experimental simplicity and physiological relevance—surpassing both enzymatic and conventional scratch assays in terms of control, reproducibility, and the capacity to integrate physical and biological variables in a single platform [1,10,62,63].

On the other hand, physical exclusion models utilize barriers, stoppers, or temporary microstructures that block cell adhesion in specific regions of the chip. After removal, cell-free areas with regular contours and without mechanical damage are formed, ensuring high geometric reproducibility [25,29,38,44,48]. This approach allows for multiple simultaneous wounds on a single device, increasing experimental throughput and standardization [25,44,48,64].

Integration with controlled laminar microflow enables the study of interactions between mechanical and chemical gradients, revealing synergistic effects between shear stress, paracrine signaling, and ECM remodeling [29,38,44,48]. It is especially advantageous in endothelial and epithelial models, whose migration depends on cell–cell communication [6,64]. The use of integrated co-cultures expands the translational potential, allowing the investigation of angiogenesis and multicellular regeneration under controlled physiological conditions [38,44,48].

Compared to the scratch assay, physical exclusion eliminates mechanical damage and reduces operator-dependent variability, being compatible with high-resolution microscopy and automated monitoring [6,64]. Its geometric precision allows for standardized quantification of migration, which is absent in the traditional method. However, the removal of the barriers can alter local gradients and reduce chemical confinement, requiring precise calibration of flow parameters [25,29,44,48,64].

From a biological point of view, the technique is indicated for studying collective migration, coordinated regeneration, and cellular cooperation, analyzing polarization, actin, and the formation of intercellular junctions [6,38,44,48,64]. Factors such as barrier width, perfusion rate, and substrate composition influence the closure speed and edge morphology. Adjustments in the microfluidic layout correlate with ECM reorganization, providing reproducible metrics for wound healing analysis [6,64]. Thus, physical exclusion represents a methodological advance over the scratch assay, by controlling the microenvironment and integrating physical and biological variables with greater precision [6,64].

Conversely, the geometry and architecture of physical depletion and exclusion methods are often integrated into more complex geometries, such as multi-chamber or radial layouts, pillar-based patterns, or detachable molds, that enable the simultaneous generation of multiple wounds with distinct shapes and sizes [38,44,65]. This geometric versatility enhances analytical capacity and throughput but demands greater precision in fabrication and flow control, often requiring external pumps or pneumatic systems. Thus, selecting the appropriate geometry involves balancing experimental objectives, equipment availability, and desired reproducibility. Linear single-channel designs remain advantageous for controlled mechanistic assays, whereas multi-chamber or array-based systems are preferred for large-scale or automated analyses.

The assessment of cost-effectiveness among different microfluidic approaches involves both technological costs, related to device fabrication and infrastructure, and operational costs, associated with reagent consumption and experimental time. Although the initial investment in microfabrication and flow systems is higher than that of the conventional scratch assay, the per-experiment cost decreases substantially through device reuse and optimization [25,41]. The intrinsic miniaturization of microfluidic platforms reduces reagent consumption by up to 90%, resulting in significant savings and higher experimental throughput [9,25]. Passive-flow devices, based on gravity or capillary effects, eliminate the need for pumps and valves, lowering operational costs by up to 70% compared with active systems [42,46]. Despite a higher initial cost, the ability to run multiple conditions within a single chip and the use of minimal volumes provide superior cost–benefit efficiency [41]. In contrast, the scratch assay accumulates indirect costs due to operator-dependent variability and the need for multiple replicates [7].

Regarding the biological models used, our review found that the choice of cell type was not directly linked to the wound-creation methodology but rather to the overarching biological question addressed. Tumor cell lines are widely used in scratch assays investigating cancer-related migration and metastasis, supporting the evaluation of anti-metastatic agents [66]. Conversely, fibroblasts and keratinocytes are preferred for skin-healing studies, as they reproduce the physiological dynamics of wound closure and enable the assessment of therapeutic agents such as cytokines or growth factors [67]. Endothelial cells, particularly HUVECs, are commonly employed in angiogenesis studies investigating responses to VEGF gradients [68].Therefore, while enzymatic depletion in simple geometries is well suited for mechanistic migration studies, physical or exclusion-based wound-creation methods combined with multicellular or 3D configurations may be more appropriate for translational research and drug-response profiling. Selecting the appropriate combination of cell model, wound strategy, and chip complexity is thus essential for aligning the experimental design with the intended clinical or pharmacological objective.

Regardless of the methodology employed for wound formation, more than half of the studies reported the use of at least one type of ECM [25,30,32,33,34,35,37,38,41,42,43,44,45,49]. This choice is justified by the fact that the ECM plays a fundamental role not only as a physical substrate for cell adhesion but also as a modulator of signaling pathways, directly influencing the migration process [45]. Different ECM compositions, with variations in stiffness and density, can alter both the speed and the pattern of cell movement, which reinforces its importance for models that more closely mimic the in vivo microenvironment, even though it may introduce greater variability in the results [11,69]. Interestingly, only one study used the ECM not as a coating, but as a method to generate three-dimensional cultures [27]. This finding is relevant because, although 3D culture makes the assay more physiological, it also brings technical challenges, such as increased complexity in analysis and difficulties in standardization, which may explain the limited adoption of this strategy in the reviewed works [70].

Building on these findings, microfluidic-based wound healing models emerge as a versatile and promising platform with significant translational potential, particularly for applications in personalized medicine and mechanobiology. By enabling precise control over the wound microenvironment, including shear stress, chemical gradients, and ECM interactions, these systems achieve a level of physiological fidelity unattainable in conventional assays. Moreover, the use of patient-derived primary cells or biopsies allows the recreation of individualized responses to therapeutic interventions [7,25,31,39], supporting the study of complex and patient-specific pathophysiological conditions such as chronic or diabetic wounds, often modeled through functional bioelectronic or electroactive devices [36].

The diversity of objectives identified across the reviewed studies further illustrates the broad applicability of microfluidic wound healing assays, ranging from drug testing to bioengineering approaches. This methodological flexibility, combined with the variety of imaging techniques, quantification tools, and observation times reported, reinforces the adaptability of these platforms to experiments with different levels of complexity. Despite this versatility, the translation of microfluidic wound healing systems into clinical and regulatory contexts remains limited. The main barriers lie in the absence of standardized validation protocols, the high variability in device designs, and the lack of harmonized guidelines for lab-on-a-chip technologies. Establishing reproducible quality-control criteria and performance benchmarks, comparable to those used for medical devices and diagnostic tools, will be essential for advancing their clinical adoption [11,31].

Although microfluidic devices present some practical and technical limitations, such as the requirement for specialized equipment and the complexity of chip fabrication, these challenges do not diminish their potential [71]. On the contrary, they highlight the need for future advances aimed at simplifying both design and experimental protocols, thereby increasing the accessibility of this technology [62]. Ensuring a stable microenvironment and maintaining precise control of gradients and flows remain challenges but overcoming them will make these assays even more robust and broadly applicable [62,71].

Our findings further emphasize that each methodology offers complementary strengths: enzymatic cell depletion preserves the ECM, enabling studies of cell–ECM interactions in conditions closer to the in vivo environment; in contrast, physical depletion and exclusion strategies support more automated and scalable assays, well suited for high-throughput applications [12]. Moreover, the inherent miniaturization of microfluidic systems reduces experimental costs by lowering sample and reagent consumption [72,73]. Taken together, these results demonstrate that scratch assays in lab-on-a-chip platforms are not only feasible and reproducible but also represent a promising alternative to conventional methods, aligned with current demands for standardization, automation, and efficiency in biomedical research.

This review presents some limitations, such as the lack of a quantitative meta-analysis, due to the variability of the reported parameters, such as migration rate, wound closure percentage, and shear stress intensity. A systematic economic analysis was also not conducted, which restricts the objective comparison of costs and efficiency between the microfluidic models. Furthermore, there was no classification of technical performance regarding reproducibility, stability, automation, and cost per assay.

## 5. Conclusions

This systematic review demonstrates that microfluidic models applied to wound healing assays represent a paradigmatic transition from traditional two-dimensional experimentation to physiologically relevant, dynamic, and quantitative platforms. Wound healing assays performed in microfluidic devices have proven to be a promising alternative to conventional methods, offering greater reproducibility, improved microenvironmental control, and experimental conditions that more closely mimic the in vivo context.

Among the methodologies evaluated, enzymatic depletion stands out for preserving the ECM and enabling the precise manipulation of chemical gradients, while physical depletion and exclusion approaches are better suited for high-throughput assays, supporting multiple geometries and greater standardization. The incorporation of additional factors—such as different ECM compositions, co-cultures, and shear stress—further expands the applicability of these platforms for studying wound healing, tumor progression, and therapeutic screening.

Based on the comparative analysis, several practical recommendations can guide the design of microfluidic scratch assays. Device architecture should ensure consistent wound definition and ECM preservation, prioritizing non-destructive wound-generation methods when higher physiological fidelity is required. Microchannels should support stable chemical and mechanical gradients while remaining compatible with routine imaging and operation. Material selection must balance biocompatibility, optical clarity, and scalability, and culture conditions, such as flow rate and shear stress, should be aligned with the biological model. Finally, cost–benefit considerations are essential, with simpler exclusion-based devices suitable for routine assays and more complex flow-controlled platforms recommended for applications requiring enhanced physiological relevance.

From a technological perspective, advances in microfabrication, miniaturization, and pump-free design have reduced costs and made these systems more accessible, sustainable, and compatible with automated analyses. In parallel, the integration of embedded sensors and artificial intelligence holds the potential to standardize measurements, increase precision, and reduce animal use in research.

Despite persistent challenges related to device fabrication and the lack of interlaboratory standardization, recent progress indicates a rapid consolidation of these models as versatile and cost-effective tools. In summary, microfluidic-based wound healing assays are emerging as a strategic technology for automation, reproducibility, and efficiency in biomedical research, marking an important step toward more realistic and translational experimental models.

## Figures and Tables

**Figure 1 cells-14-01931-f001:**
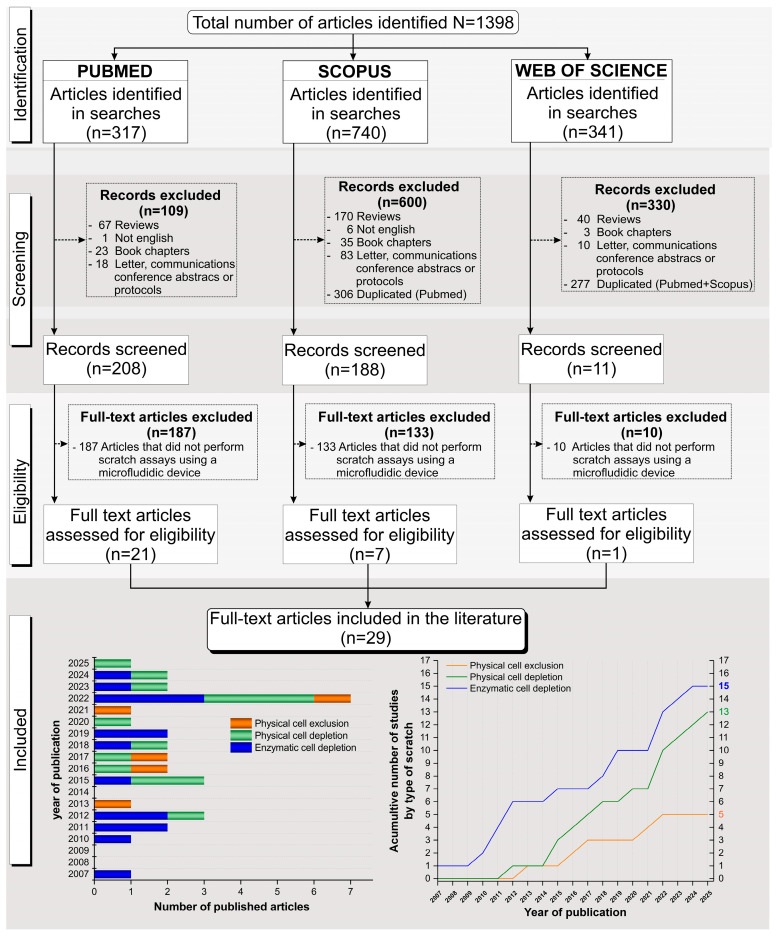
Flowchart corresponding to the PRISMA guidelines, covering the processes of identification, screening, eligibility, and inclusion of the articles selected for this review.

**Figure 2 cells-14-01931-f002:**
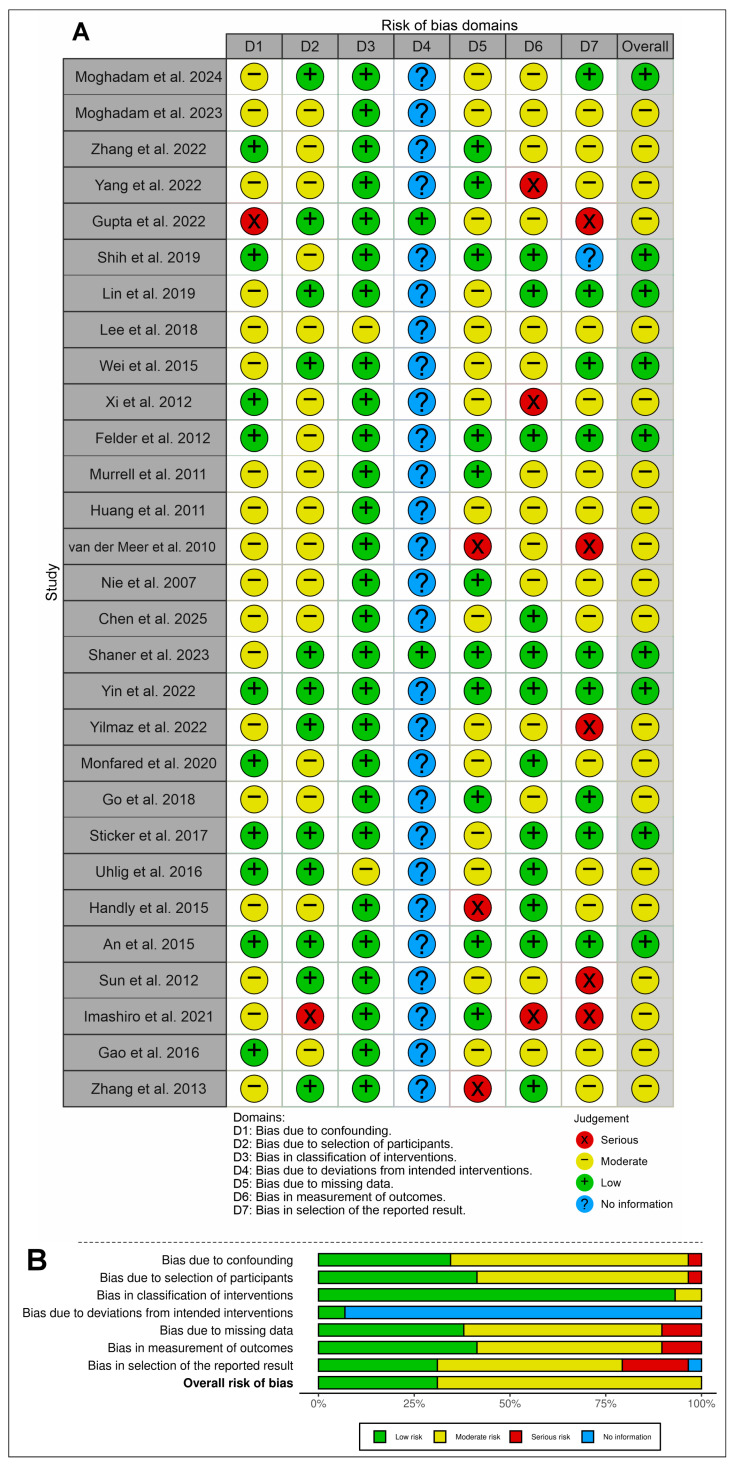
Risk of bias assessment of the included studies using the ROBINS-I. (**A**) Visual representation of the domain-level risk of bias judgments (D1 to D7) for each study. Each cell indicates the assigned risk level based on consensus among reviewers: green (+) for low risk, yellow (−) for moderate risk, and red (×) for serious risk. The “Overall” column reflects the overall risk of bias rating across all seven domains [8,9,24,25,26,27,28,29,30,31,32,33,34,35,36,37,38,39,40,41,42,43,44,45,46,47,48,49,50]. (**B**) Summary of the proportion of risk of bias judgments across all included studies for each domain. Bars depict the distribution of low, moderate, no information, and serious risk classifications.

**Figure 3 cells-14-01931-f003:**
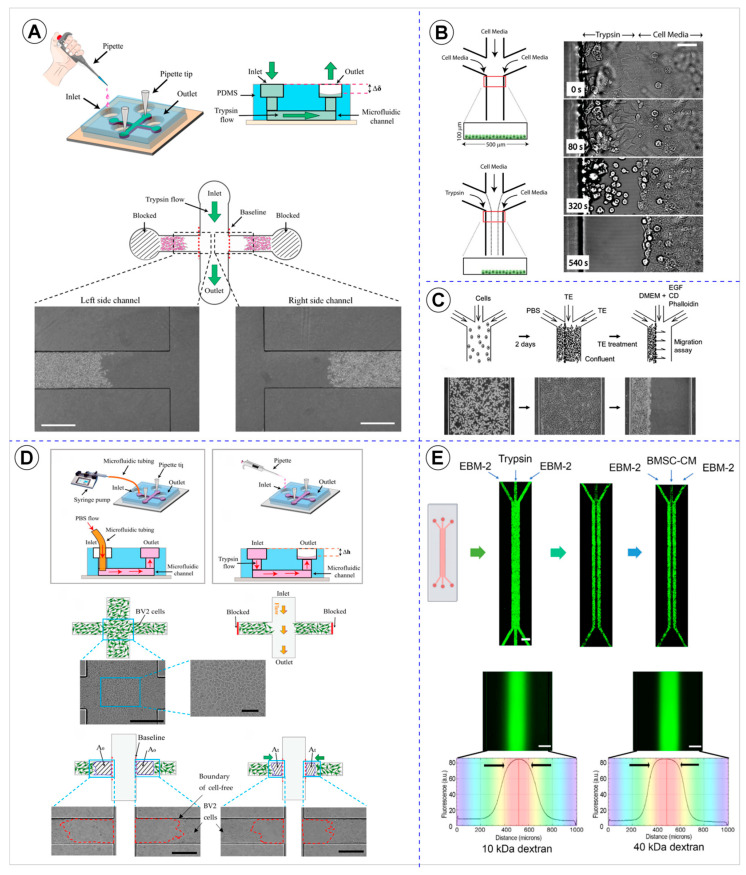
Representative examples of microfluidic-based wound healing assays with wound formation by enzymatic cell depletion. (**A**) A defined cell-free wound was created by gravity-driven trypsin flow in a central microchannel to assess BV2 microglial cell migration on various ECM substrates (adapted from [45]). (**B**) Multiple laminar flows were used to selectively cleave cells enzymatically, creating a “damage-free” denudation. This method separated the influence of free space from cell injury to study the collective migration of an epithelial sheet (adapted from [32]). (**C**) A laminar trypsin flow within microchannels was used to achieve well-controlled cell detachment, patterning precise wound edges in a confluent monolayer. This method enables an accurate cell migration assay with minimal reagent consumption (adapted from [34]). (**D**) A bi-functional microfluidic chip generated cell-free wounds using either trypsin (chemical) or PBS flow (mechanical). This approach enabled direct comparison of microglial BV2 cell migration following different wounding methods within the same device (adapted from [46]). (**E**) A microfluidic device employed laminar trypsin flow to create a defined wound, enabling the study of endothelial cell migration under combined biochemical (BMSC-conditioned medium) and biomechanical (fluidic shear stress) stimuli (adapted from [49]).

**Figure 4 cells-14-01931-f004:**
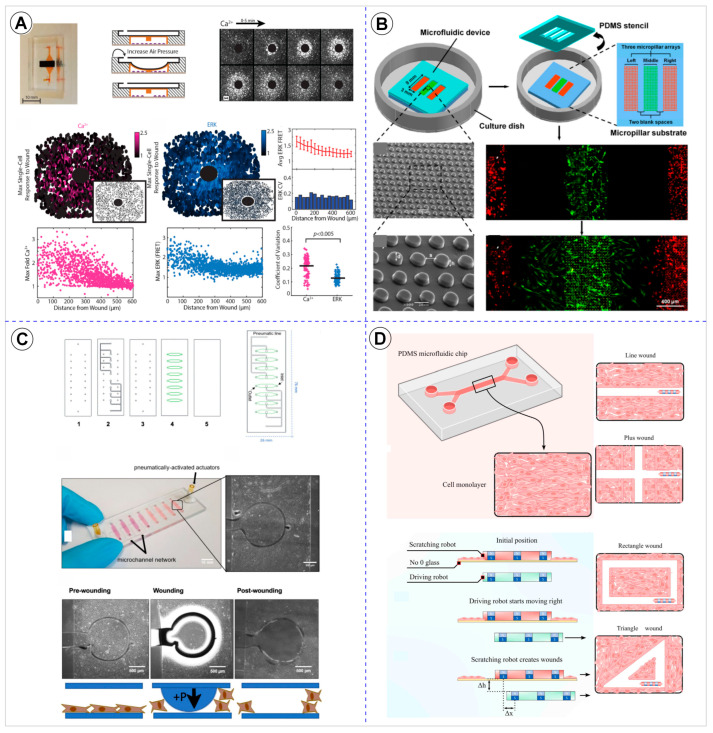
Representative examples of microfluidic-based wound healing assays with wound formation by physical cell depletion. (**A**) A microfluidic device was used to create an in vitro wound and quantify Ca^2+^ and ERK signaling gradients. The study revealed that paracrine communication optimizes gradient accuracy by balancing signal magnitude and noise reduction (adapted from [33]). (**B**) A microfluidic device featuring a tunable micropillar substrate was designed to mimic 3D extracellular matrix topographies. This platform enabled positional co-culture of three cell types and the creation of a wound to investigate collaborative cell–microenvironment interactions during healing (adapted from [24]). (**C**) A pneumatically actuated circular membrane was used to mechanically generate a highly reproducible wound on a fibroblast monolayer, automating the wounding process for the study of cell migration under various stimulatory and inhibitory conditions (adapted from [35]). (**D**) An untethered magnetic microrobot was manipulated inside a closed microfluidic channel to create wounds of uniform size and different geometries, enabling the study of how wound shape influences healing dynamics without chemical or physical intrusion (adapted from [47]).

**Figure 5 cells-14-01931-f005:**
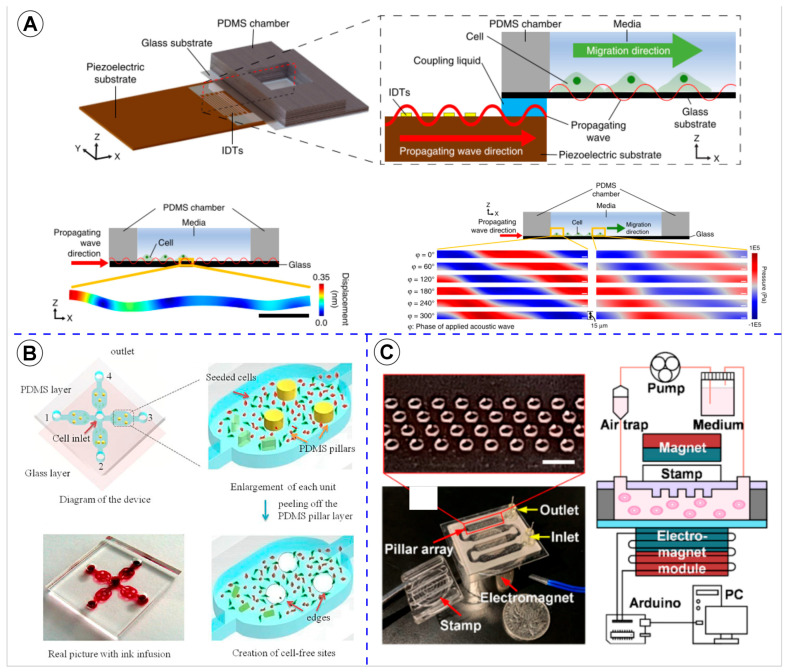
Representative examples of microfluidic-based wound healing assays with wound formation by physical cell exclusion. (**A**) Surface acoustic waves were applied to a cell culture substrate within a microfluidic device to investigate their effect on collective cell migration (adapted from [29]). (**B**) A microfluidic device used arrays of micropillars to define and create localized cell-free regions, enabling parallel quantitative investigation of cell migration and proliferation in response to biochemical stimuli like EGF (adapted from [48]). (**C**) A microfluidic device employed a novel mechanical method to simultaneously create multiple uniform cell-free zones under controlled shear stress, enabling reproducible and biomimetic migration studies for drug testing and anti-angiogenesis research (adapted from [44]).

**Figure 6 cells-14-01931-f006:**
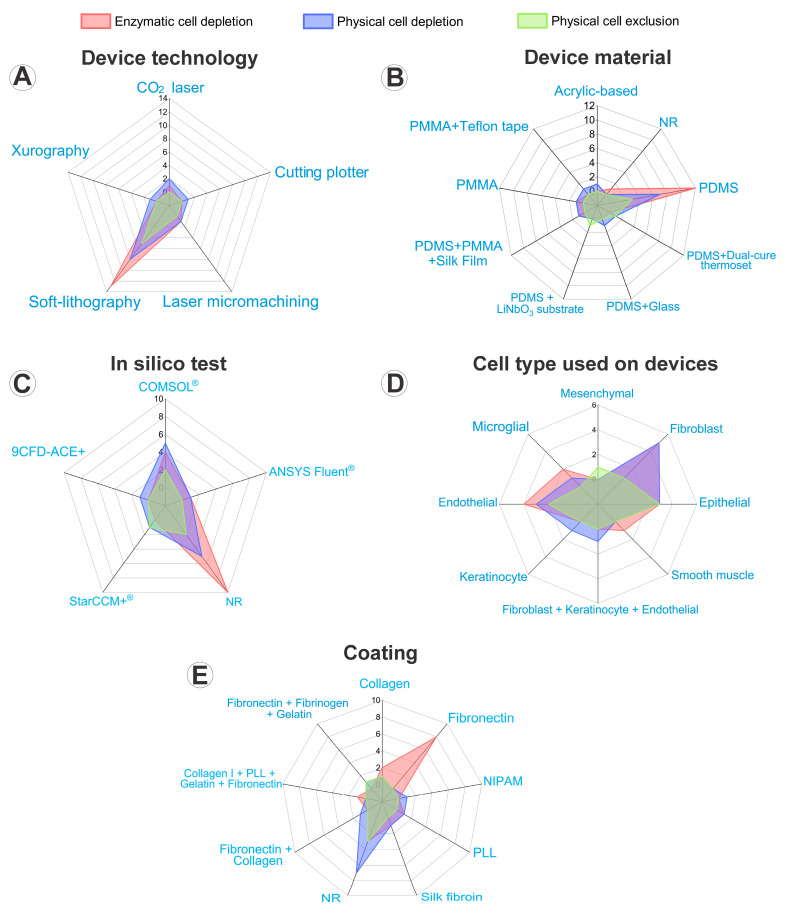
Illustrative graphs summarizing the main findings of this review, organized according to the tables. (**A**–**C**) Data from Table 1: radar charts representing (**A**) device fabrication techniques, (**B**) fabrication materials, and (**C**) use of in silico simulations. (**D**,**E**) Data from Table 2: radar charts showing (**D**) cell types used in devices and (**E**) coating strategies applied. Abbreviations: PMMA: Polymethyl methacrylate; NR: Not reported; PDMS: Polydimethylsiloxane; LiNbO3: Lithium niobate; 9CFD-ACE: Computational fluid dynamics by ACE+ software; NIPAM: N-Isopropylacrylamide; PLL: Poli-L-lysine.

**Figure 7 cells-14-01931-f007:**
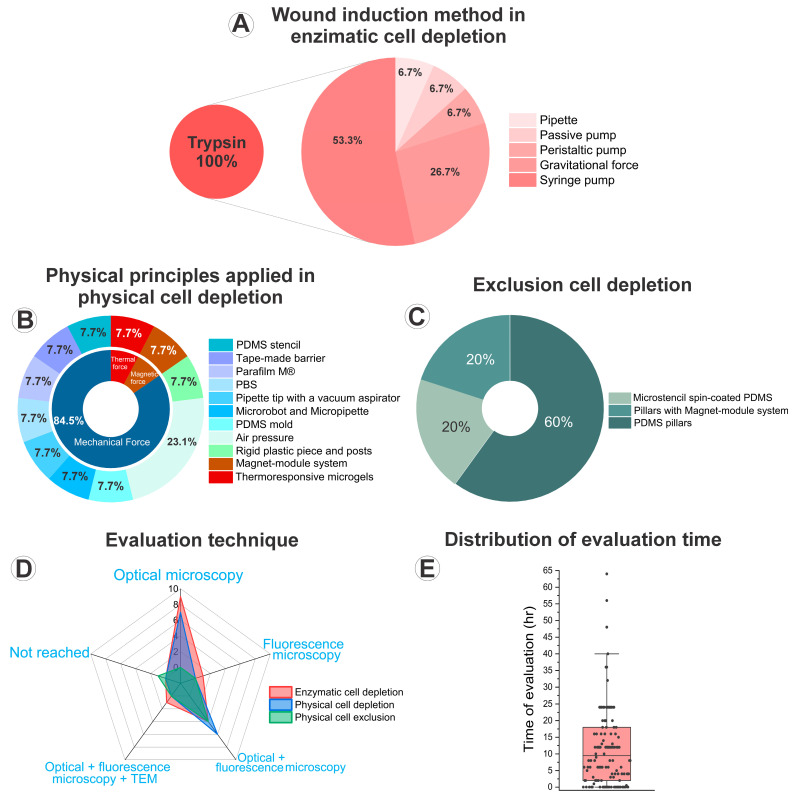
Illustrative graphs summarizing the main findings of this review, organized according to the tables. (**A**–**C**) Data from Table 3: pie charts illustrating methodology-specific aspects, including (**A**) wound induction methods in enzymatic cell depletion, (**B**) physical principles applied in physical cell depletion, and (**C**) materials used in physical cell exclusion. Percentages represent the number of articles within each methodology. (**D**,**E**) Data from Appendix A: (**D**) radar chart summarizing wound evaluation techniques and (**E**) boxplot representing the distribution of wound assessment times. In all radar charts, the three colors represent the different wound formation methodologies (enzymatic cell depletion, physical cell depletion, and physical cell exclusion). Abbreviations: PDMS: Polydimethylsiloxane; PBS: Phosphate-buffered saline; TEM: Transmission electron microscopy.

**Figure 8 cells-14-01931-f008:**
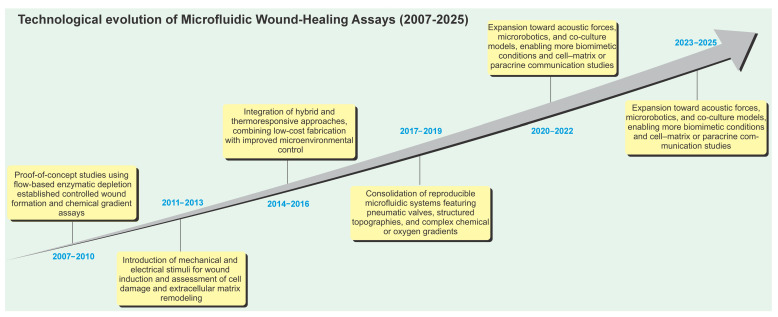
Technological evolution of microfluidic wound healing assays (2007–2025). Timeline illustrating the progressive evolution of microfluidic wound healing models, from early enzymatic depletion systems to advanced biomimetic and passive-flow platforms emphasizing accessibility and standardization.

**Figure 9 cells-14-01931-f009:**
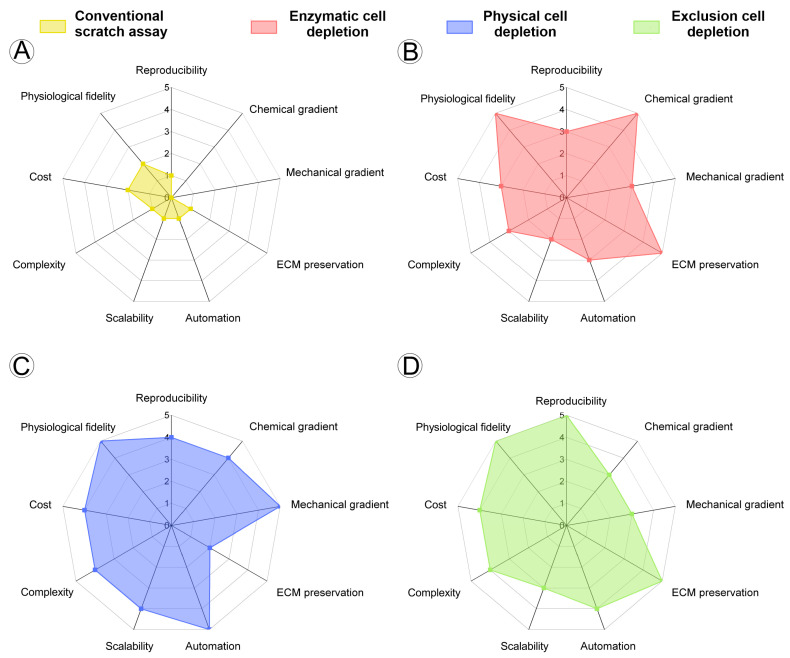
Radar plots comparing key performance parameters among different wound healing assay methodologies. (**A**) Conventional scratch assay; (**B**) Enzymatic cell depletion; (**C**) Physical cell depletion; (**D**) Exclusion cell depletion. Each plot qualitatively represents nine parameters: reproducibility, physiological fidelity, chemical and mechanical gradients, ECM preservation, automation, scalability, complexity, and cost.

**Table 1 cells-14-01931-t001:** Characteristics, design, and fabrication of microfluidic devices for scratch assays in wound healing studies.

Study	Year	Mold Cast		Microfluidic Device				Device Assembly	In Silico Test	
		Material	Fabrication	Material	Layers	Technology	Microchannels Geometry	Cover	Sealing	
Scratch assay: Enzymatic cell depletion										
Moghadam et al. [46]	2024	SU-8	Photolithography	PDMS	2	Soft-lithography	1 main channel (W = 0.9 mm, L = 8.2/8.4 mm) and 2 side channels (W = 200 or 400 µm, L = 6 mm); 2 inlets and 2 outlets (D = 1.5 mm) and 4 reservoirs (D = 8 mm)	Glass	Plasma	COMSOL^®^
Moghadam et al. [45]	2023	SU-8	Photolithography	PDMS	2	Soft-lithography	1 main channel (W = 0.9 mm, L = 5.6 mm and H = 100 µm) and 2 side channels (W = 600 µm, L = 4 mm and H = 100 µm); 4 reservoirs (D = 8 mm); 2 inlets and 2 outlets (D = 3 mm)	Glass	Plasma	NR
Zhang et al. [49]	2022	SU-8	Photolithography	PDMS	1	Soft-lithography	1 main channel (L = 150 mm, W = 1 mm, H = 100 µm); 3 short inlet and 3 outlet channels on both ends	Glass	Plasma	NR
Yang et al. [44]	2022	NR	NR	PDMS	1	Soft-lithography	1 main channel (W = 3 mm, H = 150 µm, L = 300 mm); 3 inlets and 1 outlet	Glass	Plasma	COMSOL^®^
Gupta et al. [27]	2022	3D-printed material	3D printing	PDMS, PMMA, and silk film	2 and 3	Laser micromachining	3 parallel microchannels (H = 200 µm, W = 200 µm); 3 inlets and 3 outlets (D = 3 mm)	Glass	Super glue	ANSYS Fluent^®^
Shih et al. [37]	2019	SU-8	Photolithography	PDMS	1	Soft-lithography	2 channels sets: 1 cell culture straight channel, 3 inlets and 1 outlet, and 1 mixing channel	Glass	Plasma	NR
Lin et al. [31]	2019	NA	NA	PMMA	6	CO_2_ laser	1 central inlet for trypsin flow, a side inlet for medium flow, and 1 outlet; 3 different W cell culture areas (6 mm, 4.5 mm, and 3 mm)	PMMA	Double-sided tapes	COMSOL^®^
Lee et al. [30]	2018	SU-8	Photolithography	PDMS	2	Soft-lithography	1 cell culture region (W = 500 µm, L = 10 mm, and H = 200 µm); 2 inlets and 1 outlet	Nano-patterned PDMS slab	Plasma	NR
Wei et al. [42]	2015	SU-8	Photolithography	PDMS	1	Soft-lithography	1 main channel (W = 800 µm, L = 3 mm, H = 100 µm or 250 µm); 2 inlets and 1 outlet (D = 5 mm)	Glass	NR	NR
Xi et al. [43]	2012	NR	Photolithography	PDMS	1	Soft-lithography	1 long channel (W = 900 µm, H = 60 µm, L = 3 mm) separated on 1 side into 3 smaller inlet channels (W = 200 µm, H = 100 µm, L = 2 mm)	Culture dish	NR	NR
Felder et al. [8]	2012	SU-8	Photolithography	PDMS	1	Soft-lithography	1 microwell (W = 900 µm, H = 100 µm, L = 14 mm); 2 inlet microchannels (W = 260 µm, H = 100 µm, L = 12 mm); 3 outlet microchannels (W = 260 µm, H = 100 µm, L = 6 mm)	Glass	Plasma	NR
Murrell et al. [32]	2011	NR	Photolithography	NR	1	Soft-lithography	3 channels (W = 500 µm) that converge into a single channel (W = 500 µm, L = 10 mm)	NR	NR	COMSOL^®^
Huang et al. [28]	2011	SU-8	Photolithography	PDMS	1	Soft-lithography	1 main channel (L = 1 cm, W = 900 µm, H = 60 µm); 3 inlet channels (W = 300 µm)	Glass	PDMS	NR
van der Meer et al. [41]	2010	SU-8	Photolithography	PDMS	1	Soft-lithography	1 long channel (H = 60 µm, W = 500 µm, L = 2 cm) separated on one side into 3 smaller inlet channels	Glass	Plasma	NR
Nie et al. [34]	2007	Poly(UA-co-IBA)	Photolithography	PDMS	1	Soft-lithography	3 inlet microchannels (W = 300 µm) that converged into a single main channel (W = 900 µm)	Collagen-coated culture dish	NR	NR
Scratch assay: Physical cell depletion										
Chen et al. [9]	2025	Photosensitive ink	3D printing	PDMS	7	Soft-lithography	3 channels, 2 reservoirs, and 1 cell chamber	Glass and PMMA	NR	COMSOL^®^
Moghadam et al. [46]	2024	SU-8	Photolithography	PDMS	2	Soft-lithography	1 main channel (W = 900 µm, L = 8.2/8.4 mm) and 2 side channels (W = 200 or 400 µm, L = 6 mm); 2 inlets and 2 outlets (D = 1.5 mm) and 4 reservoirs (D = 8 mm)	Glass	Plasma	COMSOL^®^
Shaner et al. [36]	2023	NA	NA	Acrylic-based double-sided pressure-sensitive adhesive	2	CO_2_ laser	1 center channel (H = 500 µm, W = 300/600 and 900 µm); 4 reservoirs (H = 8 mm); 3 inlets and 1 outlet	Culture dish	Adhesion	COMSOL^®^
Yin et al. [44]	2022	SU-8	Photolithography	PDMS	2	Soft-lithography	100 cylinders (D = 400 µm, H = 100 µm) around 3.5 mm × 20 mm areas; 4 connected chambers on sidewall (21 mm × 5 mm × 0.8 mm); 1 inlet and 1 outlet (D = 1.25 mm)	Culture dish	Plasma	COMSOL^®^
Yilmaz et al. [47]	2022	SU-8	Photolithography	PDMS	1	Soft-lithography	Square cell seeding area (L = 10 mm, W = 10 mm)	Glass	Plasma	COMSOL^®^
Gupta et al. [27]	2022	3D-printed material	3D printing	PDMS, PMMA and silk film	2 and 3	Laser micromachining	1 multiple zigzag microchannel (H = 200 µm, W = 200 µm); 1 inlet and 1 outlet (D = 3 mm)	Glass	Super glue	ANSYS Fluent^®^
Monfared et al. [35]	2020	NA	NA	PDMS and glass	4	Xurography	8 microchannels (W = 2.5 mm, L = 8.5 mm, H = 250 µm); 8 circular wound areas (1.5 mm^2^); 1 inlet and 1 outlet (D = 6 mm)	Glass	Plasma	NR
Go et al. [26]	2018	SU-8	Photolithography	PDMS	1	Soft-lithography	1 cell culture square microchannel (W = 1 cm, L = 1 cm, H = 100 µm); 6 microposts (D = 400–800 μm) situated 1050 μm away from the glass substrate; 2 inlet cells and media reservoirs (D = 8 mm)	Glass	Plasma	NR
Sticker et al. [38]	2017	TMMF S2045	Photolithography	PDMS and dual-cure thermoset	3	Soft-lithography	1 circular shaped frame for cell depletion (D = 1.5 and 2.5 mm); 4 cell culture chambers (H = 90 μm high, W = 2.5 mm, and L = 7.5 mm)	Glass	Heating	StarCCM+^®^
Uhlig et al. [40]	2016	NA	NA	PMMA	2	Cutting plotter	1 microchannel (W = 100 µm, L = 1 cm); 3 inlets	Glass	Double-sided tapes	NR
Handly et al. [33]	2015	SU-8	Photolithography	PDMS	2	Soft-lithography	1 trapezoid cell chamber (W = 4 mm, L = 4 mm, H = 60 µm); 1 air channel	Glass	NR	NR
An et al. [24]	2015	AZ 50XT	Photolithography	PDMS	2	Soft-lithography	3 micropillar arrays (W = 3 mm, L = 9 mm, H = 500 µm) with 0.8 mm space between; pillars had a H = 10 µm, and diameter of 15, 18, or 21 µm	Culture dish	PDMS	NR
Sun et al. [39]	2012	NA	NA	PMMA and Teflon tape	6	CO_2_ laser	1 central long slit (H = 320 µm) connected to 3 inlet holes (D = 8 mm), with 3 salt bridge ports (D = 5 mm), and 3 fluid ports (D = 2 mm)	Culture dish	Double-sided tapes and heating	9CFD-ACE+ (CFD-GEOM, CFD-ACE, and CFD-VIEW)
Scratch assay: Physical cell exclusion										
Yin et al. [44]	2022	SU-8	Photolithography	PDMS	2	Soft-lithography	100 cylinders (D = 400 µm, H = 100 µm) around 3.5 mm × 20 mm areas; 4 connected chambers on sidewall (21 mm × 5 mm × 0.8 mm); 1 inlet and 1 outlet (D = 1.25 mm)	Culture dish	Plasma	COMSOL^®^
Imashiro et al. [29]	2021	Photosensitive ink	3D printing	PDMS and LiNbO_3_ substrate	2	Soft-lithography	NR	Glass	Plasma	COMSOL^®^
Sticker et al. [38]	2017	TMMF S2045	Photolithography	PDMS and dual-cure thermoset	3	Soft-lithography	1 circular shaped frame for cell depletion (D = 1.5 and 2.5 mm); 4 cell culture chambers (H = 90 μm high, W = 2.5 mm, and L = 7.5 mm)	Glass	Heating	StarCCM+^®^
Gao et al. [25]	2016	Photoresist film	Photolithography	PDMS	2	Soft-lithography	1 main channel (W = 250 µm, H = 40 µm, L = 4 mm); 2 inlets and 2 outlets	Glass and PMMA	Plasma and screws	NR
Zhang et al. [48]	2013	SU-8	Photolithography	PDMS	1	Soft-lithography	4 uniform units with 3 pillars each (D = 800 µm); 1 inlet and 4 outlets	Glass	Plasma	NR

**Abbreviations:** SU-8: Negative epoxy-based photoresist; NR: Not reported; 3D: Three dimensions; NA: Not applicable; Poly(UA-co-IBA): Copolymers of 2-acrylamido-2-methylpropane sulfonic acid and isobutyl acrylate; TMMF S2045: Photosensitive epoxy laminate; AZ 50XT: Positive photoresist; PDMS: Polydimethylsiloxane; PMMA: Polymethyl methacrylate; LiNbO3: Lithium niobate; W: Width; L: Length; D: Diameter; H: Height; 9CDF-ACE+: Computational fluid dynamics by ACE+ software.

**Table 2 cells-14-01931-t002:** Cell culture in microfluidic devices.

Study	Cell Line	Cell Type	Concentration (×10^6^ Cells/mL)	Seeding Cells Method	Medium	Type of Culture	Coating	Incubation Time (h)
Scratch assay: Enzymatic cell depletion								
Moghadam et al. [46]	BV2	Microglial	~10	Syringe (hydrostatic passive)	DMEM; 10% FBS, 1% P/S	2D	NR	NR
Moghadam et al. [45]	BV2	Microglial	10	Pipette tips (hydrostatic passive)	DMEM; 10% FBS, 1% P/S	2D	Collagen I, PLL, gelatin, and fibronectin	12
Zhang et al. [49]	HUVEC	Endothelial	5	NR	α-MEM, 10% FCS	2D	Fibronectin (40 µg/mL)	24
Yang et al. [44]	HUVEC	Endothelial	20	Pipette tips (hydrostatic passive)	DMEM; 10% FBS, 1% P/S, 1% VEGF	2D	Fibronectin (100 µg/mL)	NR
Gupta et al. [27]	L929	Fibroblast	NR	NR	DMEM; 10% FBS, 1% P/S	3D	Silk fibroin (3%)	NR
Shih et al. [37]	HUVEC	Endothelial	2	NR	CC-3162	2D	Fibronectin (100 µg/mL)	24
Lin et al. [31]	NIH/3T3	Fibroblast	3	NR	DMEM; 10% FBS	2D	NR	2.5
Lee et al. [30]	NIH/3T3	Fibroblast	5–6	NR	DMEM; 10% FBS, 1% P/S	2D	Fibronectin (40 μg/mL)	NR
Wei et al. [42]	T/G HA-VSMC	Smooth muscle	5	NR	DMEM; 10% FBS, 1% P/S	2D	Fibronectin (100 µg/mL) or collagen (1000 µg/mL)	36
	HASMC							
	RASMC							
Xi et al. [43]	NIH/3T3	Fibroblast	5	NR	DMEM; 10% FBS, P (100 U/mL), S (100 μg/mL)	2D	Fibronectin (100 µg/mL)	24
Felder et al. [8]	A549	Epithelial (alveolar basal)	6	NR (hydrostatic passive)	DMEM; 10% FBS, P (100 U/mL), S (100 μg/mL)	2D	NR	48
Murrell et al. [32]	CLS-1	Epithelial	3.75cells/chip	NR	DMEM; 10% FBS, 1% P/S	2D	Fibronectin (1000 µg/mL and 200 µg/mL)	NR
Huang et al. [28]	MCF-7	Epithelial (breast cancer)	20	Pipette tips (hydrostatic passive)	RPMI-1640; 2% de FBS	2D	PLL (50 μg/mL)	NR
van der Meer et al. [41]	HUVEC	Endothelial	2	Pipette tips (hydrostatic passive)	EGM-2	2D	Fibronectin (2000 µg/mL)	NR
Nie et al. [34]	NIH/3T3	Fibroblast	5	NR	DMEM; 10% FBS, P (200 U/mL), S (200 μg/mL)	2D	Collagen	NR
Scratch assay: Physical cell depletion								
Chen et al. [9]	DU 145	Epithelial (prostate carcinoma)	1	NR	DMEM; 10% FBS, P (100 U/mL), S (100 μg/mL), and GM (100 U/mL)	2D	NR	18
Moghadam et al. [46]	BV2	Microglial	~10	Syringe (hydrostatic passive)	DMEM; 10% FBS, 1% P/S	2D	NR	NR
Shaner et al. [36]	Epidermal keratinocytes with HPV-16 E6/E7	Keratinocyte (immortalized)	4.5	NR	KGM2 with a cocktail of factors; CaCl2, N (20 μg/mL), and Kan (100 μg/mL)	2D	NR	3
Yin et al. [44]	HeLa	Epithelial (cervical adenocarcinoma)	1	Syringe (hydrostatic passive)	DMEM; 10% FBS, 1% P/S	2D	NR	6
	HUVEC	Endothelial						
Yilmaz et al. [47]	HUVEC	Endothelial	0.02 cells/chip	NR	DMEM; 10% FBS, 1% P/S	2D	NR	24
Gupta et al. [27]	L929	Fibroblast	NR	NR	DMEM; 10% FBS, 1% P/S	3D	Silk fibroin (3%)	NR
Monfared et al. [35]	HDF	Fibroblast	0.6	NR	DMEM; 10% FBS, 1% P/S	2D	Collagen I (0.09%)	72
Go et al. [26]	BALB/3T3	Fibroblast	0.1	NR	DMEM; 10% FBS, 1% P/S	2D	NR	NR
Sticker et al. [38]	GFP-HUVEC	Endothelial (GFP-tagged)	NR	Syringe (hydrostatic passive)	EGM-2, EGM-2 SingleQuots	2D	Fibronectin (10 μg/mL), fibrinogen (5 μg/mL), and gelatin (1%)	24–48
Uhlig et al. [40]	L929	Fibroblast	2	NR	DMEM HEPES (25 mM); 10% FBS, 1% P/S, and Gln (2 Mm); CHO-K1 F12; 10% FBS, 1% P/S	2D	NIPAM	24
Handly et al. [33]	MCF-10A	Epithelial	15	Needle (hydrostatic passive)	DMEM/F12; 5% HS, 1% P/S, EGF (20 ng/mL), HC (0.5 μg/mL); CTx (100 ng/mL), insulin (10 μg/mL)	2D	Fibronectin and collagen	18–24
An et al. [24]	CCC-ESF-1; HaCaT; HUVEC	Fibroblast; keratinocyte; endothelial	0.1, 0.5, 1	NR	DMEM; 10% FBS, 1% P/S MEM-EBSS; 10% FBS, 1% P/S	2D	PLL (100 µg/mL)	24
Sun et al. [39]	NIH/3T3	Fibroblast	1	Pipette tips (hydrostatic passive)	DMEM + 10% FBS	2D	NR	3
Scratch assay: Physical cell exclusion								
Yin et al. [44]	HeLa	Epithelial (cervical adenocarcinoma)	1	Syringe (hydrostatic passive)	DMEM; 10% FBS, 1% P/S	2D	NR	6
	HUVEC	Endothelial						
Imashiro et al. [29]	NIH/3T3	Fibroblast	2	NR	DMEM; 10% FBS, 1% P/S	2D	NR	8
Sticker et al. [38]	GFP-HUVEC	Endothelial (GFP-tagged)	NR	Syringe (hydrostatic passive)	EGM-2, EGM-2 SingleQuots	2D	Fibronectin (10 μg/mL), fibrinogen (5 μg/μL), and gelatin (1%)	24–48
Gao et al. [25]	MV3 BRAFV600E	Epithelial (melanoma)	1	NR	RPMI 1640; 10% FCS	2D	Collagen I (100 µg/mL)	24–48
Zhang et al. [48]	GES-1	Epithelial	10	NR	HG-DMEM; 10% FBS, P (100 U/mL), S (100 μg/mL)	2D	NR	12
	NR	Mesenchymal stem			α-MEM; 10% FBS, P (100 U/mL), S (100 μg/mL)			
	ACCM	Epithelial (adenoid cystic carcinoma)			α-MEM; 10% FBS, P (100 U/mL), S (100 μg/mL)			

**Abbreviations:** BV2: Mouse microglial cell line; HUVEC: Human umbilical vein endothelial cells; L929: Mouse fibroblast cell line; NIH/3T3: Mouse embryonic fibroblast cell line; T/G HA-VSM: Telomerase/gene-modified human aortic vascular smooth muscle cells; HASMC: Human aortic smooth muscle cells; RASMC: Rat aortic smooth muscle cells; A549: Human alveolar basal epithelial cell line; CLS-1: Human prostate cancer cell line; MCF-7: Michigan Cancer Foundation-7 breast cancer cell line; DU 125: Human prostate carcinoma cell line; HPV-16: Human papillomavirus type 16; E6/E7: Viral oncogenes E6 and E7; HeLa: Human cervical adenocarcinoma cell line; HDF: Human dermal fibroblasts; BALB/3T3: Mouse embryonic fibroblast cell line; GFP: Green fluorescent protein-expressing; MCF-10A: Non-tumorigenic human mammary epithelial cell line; CCC-ESF-1: Human esophageal fibroblast cell line; HaCaT: Immortalized human keratinocyte cell line; MV3 BRAFV600E: Human melanoma cells with Brafv600e mutation; GES-1: Human gastric epithelial cell line; NR: Not reported; ACCM: Aortic carotid artery smooth muscle cells; DMEM: Dulbecco’s Modified Eagle Medium; FBS: Fetal bovine serum; P: Penicillin; S: Streptomycin; α-MEM: Alpha Minimum Essential Medium; FCS: Fetal calf serum; VEGF: Vascular endothelial growth factor; CC-3162: EGM™-2 product code; RPMI-1640: Roswell Park Memorial Institute Medium 1640; EGM-2: Endothelial Growth Medium-2; GM: Gentamycin; KGM2: Keratinocyte Growth Medium 2; CaCl_2_: Calcium chloride; N: Nicotinamide; Kan: Kanamycin; HEPES: 4-(2-hydroxyethyl)-1-piperazineethanesulfonic acid; Gln: L-Glutamine; CHO-K1: Chinese hamster ovary K1 cells; F12: Ham’s F-12 Nutrient Mixture; HS: Horse serum; EGF: Epidermal growth factor; HC: Hydrocortisone; CTx: Cholera toxin; MEM-EBSS: Minimum Essential Medium with Earle’s Balanced Salt Solution; HG-DEMEM: High glucose DMEM; 2D: Two-dimensional; 3D: Three-dimensional; PLL: Poli-L-lysine; NIPAM: N-Isopropylacrylamide; h: Hours.

**Table 3 cells-14-01931-t003:** Wound healing assays in microfluidic devices.

Study	Applied Material	Wound Induction Method *	Microchannels Geometry	Wound Region	Wound Geometry
Scratch assay: Enzymatic cell depletion					
Moghadam et al. [46]	Trypsin	Gravitational force	1 main channel and 2 side channels; 2 inlets and 2 outlets and 4 reservoirs	Central	Linear (A = 1,400,000 or 240,000 or 120,000 µm^2^)
Moghadam et al. [45]	Trypsin	Pipette (NR)	1 main channel and 2 side channels; 4 reservoirs; 2 inlets and 2 outlets	Central	Linear (NR)
Zhang et al. [49]	Trypsin	Syringe pump (80 µL/min)	1 main channel; 3 short inlet and 3 outlet channels on both ends	Central	Linear (NR)
Yang et al. [44]	Trypsin	Passive pump with/without siphon	1 main channel; 3 inlets and 1 outlet	Central	Linear (NR)
Gupta et al. [27]	Trypsin	Syringe pump (15 µL/min)	3 parallel microchannels; 3 inlets and 3 outlets	Central	Linear (Flow 1: 619,000 mm; Flow 2: 536,350 mm; Flow 3: 881,500 mm)
Shih et al. [37]	Trypsin	Syringe pump (10 µL/min)	2 channel sets: 1 cell culture straight channel, 3 inlets and 1 outlet, and 1 mixing channel	Central	Linear (W = 300 µm)
Lin et al. [31]	Trypsin	Syringe pump (400 µL/min)	1 central inlet for trypsin flow, a side inlet for medium flow, and an outlet; 3 different widths cell culture area	Central	Linear (W = 1.42 or 0.91 or 0.46 mm)
Lee et al. [30]	Trypsin	Syringe pump (5 to 15 µL/min)	1 cell culture region; 2 inlets and 1 outlet	Central	Linear (W = 250 to 300 μm)
Wei et al. [42]	Trypsin	Gravitational force	1 main channel; 2 inlets and 1 outlet	Lateral	Linear (A = 108.1 ± 22.9 μm or 148.9 ± 20.5 μm or 108.7 ± 10.5 μm or 383.7 ± 19.9 μm)
Xi et al. [43]	Trypsin	Gravitational force	1 long channel separated on 1 side into 3 smaller inlet channels	Central	Linear (W = 300 μm)
Felder et al. [8]	Trypsin	Peristaltic pump (3.4 µL/min)	1 microwell; 2 inlets microchannels; 3 outlets microchannels	Central	Linear (W = 300 μm)
Murrell et al. [32]	Trypsin	Syringe pump (15 µL/min)	3 channels that converge into a single channel	Lateral	Linear (W = 36 μm)
Huang et al. [28]	Trypsin	Syringe pump (50 µL/min)	1 main channel; 3 inlet channels	Lateral	Linear (NR)
van der Meer et al. [41]	Trypsin	Syringe pump (5 µL/min)	1 long channel separated on one side into 3 smaller inlet channels	Central	Linear (W = 110 μm)
Nie et al. [34]	Trypsin	Gravitational force	3 inlets microchannels that converged into a single main channel	Lateral	Linear (W = 600 μm)
Scratch assay: Physical cell depletion					
Chen et al. [9]	Parafilm M^®^	Mechanical force	3 channels, 2 reservoirs, and 1 cell chamber	Central	Linear (W = 0.5 mm)
Moghadam et al. [46]	PBS	Mechanical force (syringe pump)	1 main channel and 2 side channels; 2 inlets and 2 outlets and 4 reservoirs	Central	Linear (A = 1,400,000 or 240,000 or 12,000 µm^2^)
Shaner et al. [36]	Pipette tip with a vacuum aspirator	Mechanical force	1 center channel; 4 reservoirs; 3 inlets and 1 outlet	Central	Linear (W = 0.7 mm)
Yin et al. [44]	Magnet-module system	Magnetic force	100 cylinders around areas; 4 connected chambers on sidewall; 1 inlet and 1 outlet	Central	Circular (D = 0.4 mm)
Yilmaz et al. [47]	Microrobot and micropipette	Mechanical force	Square cell seeding area	Central	Linear, square, and triangle (NR)
Gupta et al. [27]	PDMS mold	Mechanical force	1 multiple zigzag microchannel; 1 inlet and 1 outlet	Central	Square (Flow 1: 516,130 mm; Flow 2: 512,157 mm; Flow 3: 515,600 mm)
Monfared et al. [35]	Air pressure	Mechanical force	8 microchannels; 8 circular wound areas; 1 inlet and 1 outlet	Central	Circular (D = 1.4 mm)
Go et al. [26]	Rigid plastic piece and posts	Mechanical force	1 cell culture square microchannel; 6 microposts situated away from the glass substrate; 2 inlet cells and media reservoirs	Central	Circular (D = 600 to 800 μm)
Sticker et al. [38]	Air pressure	Mechanical force	1 circular shaped frame for cell depletion; 4 cell culture chambers	Central	Circular (D = 1.5 mm)
Uhlig et al. [40]	Thermoresponsive microgels	Thermal force	1 microchannel; 3 inlets	Central	Circular (D = 200 μm)
Handly et al. [33]	Air pressure	Mechanical force	1 trapezoid cell chamber; 1 air channel	Central	Circular (D = 300 μm)
An et al. [24]	PDMS stencil	Mechanical force	3 micropillar arrays with spacing, and pillars of set height and diameter	Central	Linear (800 μm)
Sun et al. [39]	Tape-made barrier	Mechanical force	1 central long slit connected to 3 inlet holes, with 3 salt bridge ports, and 3 fluid ports	NR	Linear (NR)
Scratch assay: Physical cell exclusion					
Yin et al. [44]	Pillars with magnet-module system	Pillars removed after 6 h	100 cylinders; 4 connected chambers on sidewall; 1 inlet and 1 outlet	Central	Circular (D = 0.4 mm)
Imashiro et al. [29]	PDMS pillars	Pillars removed after 8 h	1 main channel; 2 inlets and 2 outlets	Central	Linear (A = 400 μm^2^)
Sticker et al. [38]	Microstencil spin-coated PDMS	Not reached	1 circular shaped frame for cell depletion; 4 cell culture chambers	Central	Not reached
Gao et al. [25]	PDMS pillars	NR	4 uniform units with 3 pillars each; 1 inlet and 4 outlets	Central	Linear (W = 250 μm, L = 4 mm)
Zhang et al. [48]	PDMS pillars	Pillars removed after overnight culture	4 uniform units with 3 pillars each; 1 inlet and 4 outlets	Central	Circular (D = 0.8 mm)

**Abbreviations:** PBS: Phosphate-buffered saline; Parafilm M^®^: Parafilm moisture-proof; PDMS: Polydimethylsiloxane; NR: Not reported; A: Area; W: Width; D: Diameter; L: Length; H: Hours; NA: Not applicable; ***** This column broadly refers to equipment used in enzymatic depletion, physical principles applied in physical depletion, and incubation time before pillar removal in physical exclusion.

## Data Availability

The original contributions presented in the study are included in the article. Further inquiries can be directed to the corresponding author.

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
