# Peer review of "Microfluidic-Based Scratch Assays for Wound Healing Studies: A Systematic Review"

_cells, 2025, doi:10.3390/cells14241931_

Round 1

Reviewer 1 Report (Previous Reviewer 1)

Comments and Suggestions for Authors

Authors have addressed all comments. No additional comments noted.

Author Response

Thank you very much for your time and for reviewing the revised version of our manuscript. We sincerely appreciate your comments and constructive feedback, which were essential for improving the quality of our work. We are pleased to know that all points have been satisfactorily addressed.

Thank you again for your valuable contribution.

Reviewer 2 Report (New Reviewer)

Comments and Suggestions for Authors

please see attached

Author Response

REVIEWER 2

This manuscript by Oliveira et al. presents a systematic review of microfluidic-based scratch assays (“scratch-on-a-chip”) used for wound-healing research. The review is comprehensive, well-organized, and timely, as microfluidic approaches are increasingly important for improving reproducibility and physiological relevance in cell migration studies. The manuscript covers fabrication strategies, cell culture approaches, wound induction methods, and outcomes across 29 selected studies. My comments are as follows:

1.Each panel in Figures 3–5 should be briefly described either in the main text or in the figure legends.

Answer: Thank you for your comment. A brief description of each panel in Figures 3–5 has now been added in the Results section, specifically within item 3.6, to ensure that all subpanels are clearly explained for the readers.

  1. Are permissions required for all figures?

Answer: Thank you for your comments. In our revised manuscript, Figures 3, 4, and 5 include schematic adaptations based on a selection of five representative articles for each wound-generation methodology (from the total of 29 studies included in the review). For some of these sources, permissions were not required because the original articles were published under open-access licenses that allow figure reuse with proper attribution. For the studies in which permission was required, we formally requested authorization through the RightsLink platform. All permissions obtained have been included in the submission package, as required by the journal.

  1. It would be beneficial to provide a clearer motivation for why a systematic review of microfluidic scratch assays is needed and to compare this work with existing reviews on wound-on-a-chip or migration-on-a-chip systems.

Answer: Thank you for your suggestion. We have now added, at the end of the Introduction section, a clearer justification of the aim of this systematic review, explicitly outlining its motivation and clarifying how it differs from and complements previously published reviews on wound-on-a-chip and migration-on-a-chip systems.

  1. Including a brief conceptual figure comparing conventional versus microfluidic scratch assays (highlighting different cell depletion methods) would strengthen the Introduction.

Answer: Thank you for your suggestion. This comparison was also recommended by other reviewers, and in response we developed Figure 9, which presents a comparative analysis between the conventional scratch assay and the different wound-healing methodologies identified in our review (enzymatic cell depletion, physical cell depletion, and exclusion-based approaches). The comparison encompasses key aspects such as reproducibility, physiological fidelity, chemical and mechanical gradients, ECM preservation, level of automation, scalability, complexity, and overall cost.

  1. In Section 2.1, the search strings are extremely long and should be streamlined or moved to the Supplementary Materials.

Answer: Thank you for your comments. The search strings previously shown in Section 2.1 have now been moved to the Supplementary Materials and are presented in tabulated form in Table S1. This table is explicitly referenced in the main text within the same section.

  1. The tables are informative but very large. Please consider shortening them or moving some information to a Supplementary section.

Answer: Thank you for your comments. We agree that Table 4 was particularly extensive due to the detailed textual summaries of objectives, methodologies, and outcomes for each included study. Because these descriptions are essential for illustrating the high variability of applications across microfluidic wound-healing assays and could not be further shortened without loss of meaning, we have moved Table 4 to the Supplementary Material section as Table S2. The remaining tables are more concise and focused on key technical parameters, and we believe they are sufficiently streamlined to remain in the main manuscript.

  1. Common metrics (e.g., migration rate, wound closure percentage) could be summarized and compared across studies.

Answer: Thank you for your comments. As detailed in all Tables, the substantial heterogeneity among studies, including differences in wound-generation methodologies, microfluidic device architectures, cell types, experimental conditions, and quantification strategies, resulted in highly variable metrics and reporting formats. This variability prevented the extraction of standardized parameters (such as migration rate or wound closure percentage) in a form that would allow meaningful or comparable quantitative synthesis across studies, and this information was highlighted in the discussion section.

  1. In the Conclusions, it would be useful to provide practical recommendations for designing microfluidic scratch assays.

Answer: Thank you for your suggestion. We added the practical recommendations for designing microfluidic scratch assays in the conclusion section of the manuscript.

  1. Please ensure the use of consistent units (e.g., mm, μm) across all tables.

Answer: Thank you for your comments. We conducted a full consistency check across all tables and updated the units to ensure uniform use of mm and µm throughout the manuscript, as requested.

  1. Some minor writing issues: Line 49: “biomedical”; Table 1: “CO₂”; Table 2: use “×” instead of “*”.

Answer: Thank you for your comments. All minor writing corrections requested have been implemented as suggested.

Reviewer 3 Report (New Reviewer)

Comments and Suggestions for Authors

Wound healing or wound scratch assay is an important cell biology tool used to characterize cell motility. The authors employed library and information science methodologies to survey, collect, and analyze the scope of studies, working principles, and biological contexts of microfluidic-based scratch assays. Authors systematically discussed microfluidic-based scratch assays in terms of the materials used to fabricate the devices, the mechanisms for creating scratches, such as mechanical or enzymatic removal of cell sheets, and the biological nature of the cells involved, including tumor cells, skin cells, and others. Additionally, they explored the biological significance of scratch assays under flow conditions. The review is logically structured, and sophisticated tables have been thoughtfully designed to effectively present the information. Overall, this is a well-organized and clearly presented work that is suitable for publication in its current form.

Author Response

Thank you very much for your positive and thoughtful assessment of our work. We truly appreciate your constructive feedback and are pleased that you found the manuscript well-organized and suitable for publication. Thank you again for your time and valuable contribution.

Round 2

Reviewer 2 Report (New Reviewer)

Comments and Suggestions for Authors

The revised version has been greatly improved.

This manuscript is a resubmission of an earlier submission. The following is a list of the peer review reports and author responses from that submission.

Round 1

Reviewer 1 Report

Comments and Suggestions for Authors

The authors  present here a systematic review that covers  microfluidic-based scratch assays as alternatives to conventional wound healing studies.  A range of alternative wound-induction assays are discussed which include, enzymatic cell depletion , physical cell depletion (using mechanical, thermal, magnetic forces, and physical cell exclusion, foe example  Significant advantages of microfluidic-based methods over traditional methods were presented, including enhanced reproducibility, preservation of extracellular matrix integrity,  and miniaturization with reduced reagent consumption. Overall this review article was comprehensive  and informative. Comments are as follow;

This reviewer found the organization into detailed tables examining device fabrication, cell culture methods, wound healing protocols, and outcomes is highly informative and accessible.

Inclusion of specific technical details (device dimensions, cell concentrations, flow rates) makes the review practically useful for replication.

The discussion effectively contextualizes findings within the broader field of wound healing research and identifies future research directions.

The cost-effectiveness comparisons between the different microfluidic approaches were not adequately discussed. This is important for researchers interested in implementing such methods.

It would be helpful to evaluate which cell types or tissues and research questions are best suited to each methodology.

Most, if not all included studies used custom-fabricated devices rather than commercial systems, which may limit generalizability and accessibility for some laboratories.

This review could be improved by including a discussion on the translational potential and regulatory considerations for clinical applications.

The font and graphics in most of the figures are much too small and in some cases distorted beyond recognition. For example the text and some of the illustrations in figures 2 and 3 cannot be seen even when zooming in at maximum magnification. This reviewer suggests decreasing some of the images or potentially breaking some of the figures into two figures. It’s very difficult to see anything in these images.

Reviewer 2 Report

Comments and Suggestions for Authors

The review article is highly relevant, comprehensive and rigorous, providing an extensive summary of microfluidic scratch assays. It is scientifically sound and well-structured. However, it would benefit from a few minor revisions.

  • The introduction should emphasize the novelty of the article more explicitly compared to existing reviews on microfluidic wound models.
  • The authors should consider adding a short section or paragraph highlighting the technological evolution over time from 2007 to 2025. It will help demonstrate the innovation patterns with time.
  • The review includes a lot of studies, but at times it feels more like a list of results rather than a critical analysis. It would be helpful if the authors added more discussion on what specific design features most improve reproducibility or accuracy in measuring cell movement. A short comparative insight and critical analysis could make the paper stronger. For example, maybe by saying that which setups are ideal for studying fibroblast migration and which for endothelial barrier repair.
  • The authors should consider adding a schematic contrasting the conventional scratch assay vs. microfluidic scratch-on-a-chip concept.